# Stress deficits in reward behaviour are associated with and replicated by dysregulated amygdala-nucleus accumbens pathway function in mice

Lorraine Madur[1,2,7], Christian Ineichen [1,7], Giorgio Bergamini[1], Alexandra Greter[1], Giulia Poggi [1], Nagiua Cuomo-Haymour[1,2], Hannes Sigrist[1], Yaroslav Sych[3], Jean-Charles Paterna[4], Klaus D. Bornemann[5], Coralie Viollet[6], Francesc Fernandez-Albert[6], Gregorio Alanis-Lobato[6], Bastian Hengerer [5] & Christopher R. Pryce [1,2✉]

Reduced reward interest/learning and reward-to-effort valuation are distinct, common symptoms in neuropsychiatric disorders for which chronic stress is a major aetiological factor. Glutamate neurons in basal amygdala (BA) project to various regions including nucleus accumbens (NAc). The BA-NAc neural pathway is activated by reward and aversion, with many neurons being monovalent. In adult male mice, chronic social stress (CSS) leads to reduced discriminative reward learning (DRL) associated with decreased BA-NAc activity, and to reduced reward-to-effort valuation (REV) associated, in contrast, with increased BA-NAc activity. Chronic tetanus toxin BA-NAc inhibition replicates the CSS-DRL effect and causes a mild REV reduction, whilst chronic DREADDs BA-NAc activation replicates the CSS effect on REV without affecting DRL. This study provides evidence that stress disruption of reward processing involves the BA-NAc neural pathway; the bi-directional effects implicate opposite activity changes in reward (learning) neurons and aversion (effort) neurons in the BA-NAc pathway following chronic stress.

[1] Preclinical Laboratory, Department of Psychiatry, Psychotherapy and Psychosomatics, Psychiatric University Hospital Zürich (PUK) and University of Zurich (UZH), Zurich, Switzerland. [2] Zurich Neuroscience Center, University of Zurich and ETH Zurich, Zurich, Switzerland. [3] Institute of Cellular and Integrative Neuroscience, University of Strasbourg, Strasbourg, France. [4] Viral Vector Facility, University of Zurich and ETH Zurich, Zurich, Switzerland. [5] CNS Diseases Research, Boehringer Ingelheim Pharma GmbH & Co. KG, Biberach, Germany. [6] Global Computational Biology and Digital Sciences, Boehringer Ingelheim Pharma GmbH & Co. KG, Biberach, Germany. [7] These authors contributed equally: Lorraine Madur, Christian Ineichen. ✉email: christopher.pryce@bli.uzh.ch

In mammals, distinct aversive events—stressors—stimulate changes in neural circuits that underlie adaptive brain–behaviour functions, such as emotional learning and memory leading to passive or active responding to aversion[1,2]. In contrast, chronic aversion exposure can lead to fundamentally different changes in neural circuits that in turn lead to maladaptive emotional processing and behaviour[3,4]. For example, in hippocampus and prefrontal cortex, chronic aversion leads to atrophy of dendrites of glutamate neurons, contributing to altered learning-memory for and behavioural responding to emotional stimuli[3]. In humans, chronic stress—principally chronic psychosocial stress—is recognized as a major aetiological factor for neuropsychiatric disorders including major depressive disorder (MDD) and schizophrenia[5–7]. The neural circuits and pathophysiological changes thereof that mediate between chronic aversion processing and the emergence of specific symptoms remain poorly understood. Reward processing can be markedly attenuated in such disorders, expressed behaviourally as decreases in interest and learning, and in reward-to-effort valuation (apathy), with respect to daily life events[8]. That chronic aversion processing can lead to altered reward processing indicates major crosstalk between neural circuits of aversion and reward processing. Given that the amygdala is a brain region important in both aversion and reward stimulus processing, it could be a major node in this regard[9].

The cortex-like basal nucleus of amygdala (BA) comprises primarily pyramidal glutamatergic neurons, including such with long-range axons to various cortical and sub-cortical efferent regions[10]. There is growing evidence that BA contributes to distinct neural circuits of aversion and reward processing[9]. One major projection region of BA glutamate neurons is the nucleus accumbens (NAc), which comprises primarily GABAergic neurons in its shell and core and is another major node of reward and aversion processing[11]. Recent mouse evidence indicates that BA-NAc glutamate neurons are excited by either reward or aversion: many such BA-NAc neurons are situated at the rostral-caudal intermediate BA[12,13], where the anterior and posterior BA sub-regions overlap[14]. Using in vivo single-cell recording[12] or putative genetic markers[15,16], the majority of BA-NAc neurons that were emotional-stimulus responsive were excited by either reward (sucrose, female) or aversion (quinine, foot shock); more neurons were reward- than aversion-responsive, with a minority responsive to both. Behaviourally, mice acquired operant responses for reinforcement as photostimulation of BA-NAc cell bodies; they responded at a moderate rate, consistent with the incorporation of at least some of these neurons into a reward neural pathway[17]. When a gene (Rspo2, R-spondin 2) specific to aversion-responsive BA neurons[17] was used to photostimulate BA-NAc neurons, mice displayed aversion conditioning to context and no acquisition of operant self-photostimulation, consistent with the incorporation of these neurons into an aversion neural pathway[15].

In rodents, various environmental stressors (e.g. chronic unpredictable mild stress[18], physical restraint[19], social subordinancy[20]) have been applied to investigate the effects of chronic aversion on subsequent reward and aversion processing, and associated changes in brain regions including the amygdala. For example, in mice, repeated restraint stress leads to dendritic hypertrophy of and increased excitatory neurotransmission by BA glutamate neurons[19]. Also in mice, chronic social stress (CSS) increases and decreases the salience of aversion[21] and reward[22], respectively, and leads to changes in resting-state metabolic and functional connectivity status of the amygdala in vivo[23], as well as in the transcriptome of amygdala tissue[21]. In conditioned behavioural tests, CSS mice display reduced discriminative reward learning-memory (DRLM) for a tone stimulus that predicts sucrose availability, and reduced operant responding for sucrose on an increasingly effortful reinforcement schedule (reward-to-effort valuation, REV)[22,24,25]. Such behavioural measures align with the positive valence constructs of reward responsiveness, learning and valuation, as proposed in the NIMH framework, research domain criteria (RDoC)[26]. We applied this mouse model of CSS-induced disrupted reward processing to investigate the involvement of BA-NAc glutamate neurons. Calcium-sensor fibre photometry was used to measure BA-NAc neuron activity in CSS and control mice during engagement in the reward tests. In CSS mice, the reductions in DRLM and REV co-occurred with decreased and increased BA-NAc neuron activity, respectively. To investigate for evidence of BA-NAc neuron activity-behaviour causality: when tetanus toxin light chain expression was used for chronic inhibition of the BA-NAc pathway, mice displayed CSS-like reduced DRLM and mildly reduced REV, whereas when DREADDs were used for chronic pathway activation, mice displayed intact DRLM and CSS-like reduced REV. These iterative findings demonstrate the importance of BA-NAc neurons—quite possibly separate populations of reward and aversion neurons—in mediating chronic stress effects on specific aspects of reward learning and valuation.

## Results

**Chronic social stress reduces reward learning and effort.** In chronic social stress (CSS), the CSS mouse is placed in the cage of an unfamiliar, larger, aggressive resident mouse for 30-60 s of attack, followed by physical separation and continuous sensory exposure for the next 24 h, and this is repeated with different resident mice for 15 days[21]. In this study, the mean duration of daily attack experienced by CSS mice was 45–50 s; all CSS mice displayed submissive behaviour and vocalization during the proximal stressor. In the comparison condition, mice remain in littermate pairs and are handled daily (control, CON). The first experiment demonstrated CSS effects on specific aspects of reward processing (Fig. 1a, b), replicating the previous findings[22]. Mice underwent conditioning on behaviour-outcome contingencies and were then allocated to CSS or CON. After a 1-day interval mice were studied in a discriminative reward learning-memory (DRLM) test on 3 consecutive days followed by a reward-to-effort valuation (REV) test on each of the next 2 days, with a chocolate-flavoured sucrose pellet as reinforcer. Mice received normal diet in the home cage sufficient to maintain 95-100% baseline body weight (supplementary Table S1), so that the major incentive of chocolate-flavoured sucrose was gustatory reward and not physiological satiety[22]. In the DRLM test (Fig. 1c) an initially neutral tone indicated the time-period (maximum 30 s per trial) within which a feeder response triggered sucrose pellet delivery into the feeder; such trials were separated by variable intertrial intervals (ITIs, mean 50 s, range 20–80 s) when responses were without consequence. That is, the tone is a discriminative stimulus (DS) and the reduction of DS response latency relative to the average interval between ITI responses provides a measure of reward interest/learning. Each test comprised 40 trials: across the 3 tests, the number of trials with DS feeder response was lower in CSS mice ($18.4 \pm 5.6$ per test, mean $\pm$ SD) than CON mice ($24.4 \pm 4.4$ per test) ($p = 0.01$, supplementary Table S2). Given that average DS response latency was lowest in trials 1-20 and then began to increase, as did the number of trials without a response, data analysis was conducted for trials 1–20. CSS mice made fewer DS feeder responses and therefore obtained fewer rewards than CON mice (Fig. 1d). CSS mice had longer DS response latencies than CON mice (Fig. 1e; DS duration = 30 s was used in no-response trials), and mean ITI response intervals were also longer in CSS than CON mice (Fig. 1f). For each test and mouse, an average learning ratio was calculated using per trial mean ITI response interval/DS response latency: as expected, this value was close to 1 in both groups in test 1; however, whilst it increased across tests in most CON mice

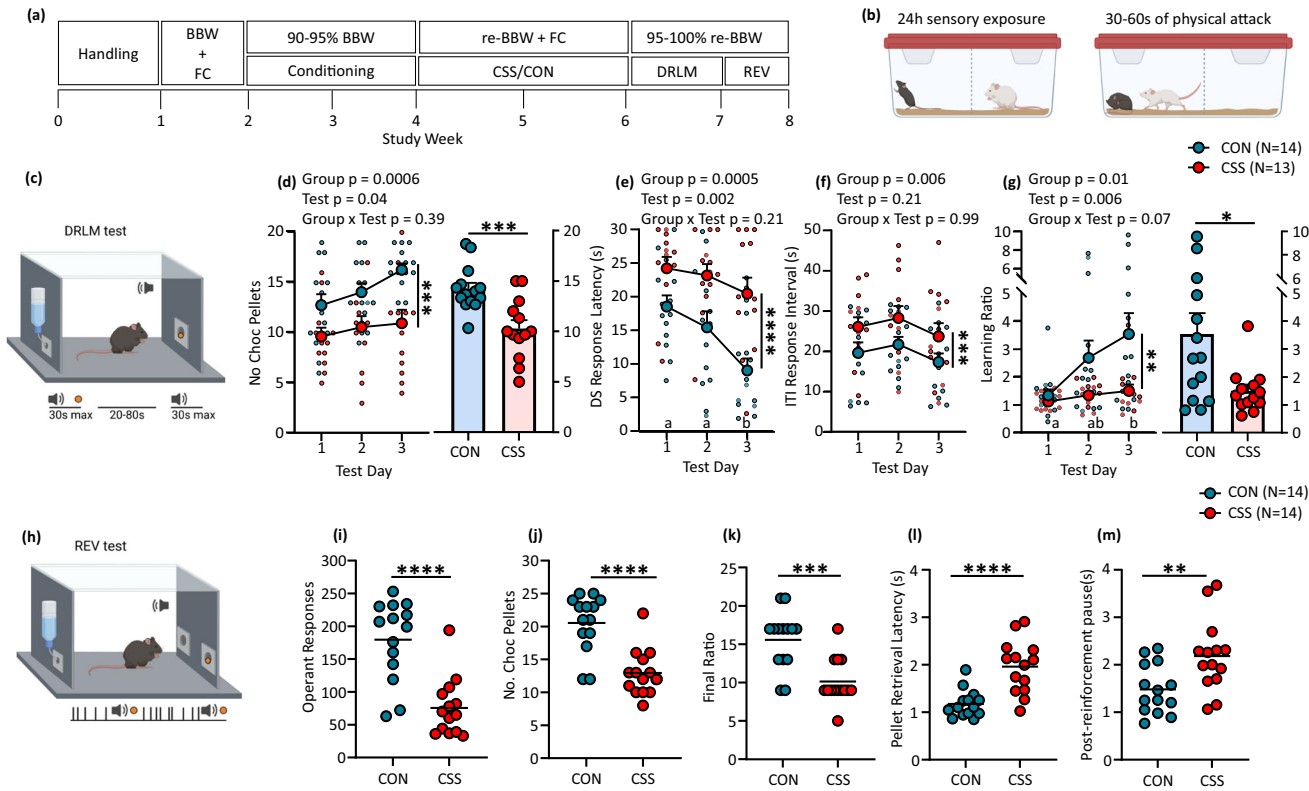

**Fig. 1 Effects of chronic social stress (CSS) on reward learning and effort. a** Experimental design. BBW + FC: measurement of baseline body weight and food consumption; 90–95% BBW/Conditioning: conditioning under food restriction that reduced BW to 90–95% BBW; CSS/CON: CSS protocol or control handling; re-BBW + FC: BW and food consumption under ad libitum feeding on days 1–12 of CSS/CON provided re-baseline values; 95–100% re-BBW: mice were mildly food restricted to be tested at 95–100% re-BBW. **b** CSS mice were placed in the cage of a dominant, aggressive CD-1 mouse to receive a 30–60 s physical attack followed by 24 h sensory exposure through a divider; this was repeated with a different CD-1 mouse on each of 15 days. CON mice were kept in littermate pairs and were handled for 1 min on each of 15 days. **c–g** Discriminative reward learning-memory (DRLM) test. **c** Tone discriminative stimulus (DS) signalled chocolate sucrose pellet (gustatory reward) availability following a feeder response; maximum DS duration was 30 s per trial and inter-trial intervals (ITIs) were 20–80 s (mean = 50 s). Mice received 3 daily tests of 40 trials each and trials 1–20 per test were used for data analysis. Data are shown as mean + SEM per test and per mouse scores. Statistical analysis was conducted using 2-way mixed-model ANOVA; 1 CSS mouse was a low-responder outlier and excluded from the analysis. **d** Number of chocolate pellets obtained, i.e. DS trials with a response, across tests (left) and individual and overall mean scores (right). **e** Median DS response latency. **f** Median ITI response interval. **g** Median learning ratio (ITI response interval/DS response latency), across tests (left) and individual median scores for test 3 (right). Test days indicated by different letters were significantly different in Tukey's multiple comparisons test: a vs. b p < 0.05 or lower. **h–m** Reward-to-effort valuation (REV) test. Data are shown as individual values and group mean values. Statistical analysis was conducted using t-tests. **h** Nose-poke responses at an operant stimulus triggered chocolate sucrose pellet delivery on a progressive ratio (PR) schedule (5 trials at PR1, 5 × PR5, 5 × PR9, 5 × PR13, etc.), signalled by a 1 s tone DS. Mice received 2 daily tests and data for test 2 are shown, as individual scores and group means; normal food was provided as a low-reward/low-effort choice. Data analysis was conducted using unpaired t-tests. **i** Number of operant responses. **j** Number of chocolate pellets earned. **k** Final ratio attained. **l** Latency to retrieve pellet after completion of ratio. **m** Post-reinforcement pause to resume operant responding. Images b, c and h were created with BioRender.com.

—primarily due to the decrease in DS response latency across tests—it remained close to 1 in most CSS mice (Fig. 1g). The next day an operant nose-poke stimulus was introduced into the test chamber, and mice underwent REV testing (Fig. 1h): the amount of reinforcement obtained was now directly dependent on the rate of operant responding, and under the condition that the number of responses required per reward, i.e. effort, increased progressively with reinforcement, thereby introducing an aversive component to the test. Reaching the required ratio resulted in a 1-s tone DS and sucrose pellet delivery. The test was conducted on 2 consecutive days, whereby the first test was used to allow adjustment to the new test conditions. On test day 2, a pellet of normal food was provided as a low-reward/low-effort choice, and data analysis was conducted for this test. CSS mice completed fewer operant responses (Fig. 1i) and consequently earned less sucrose rewards (Fig. 1j) and attained a lower final progressive ratio (Fig. 1k), than CON mice. The latency to sucrose pellet

retrieval was longer in CSS than CON mice (Fig. 1l) as was their post-reinforcement pause (Fig. 1m). Both CSS mice (0.10 ± 0.09 g) and CON mice (0.11 ± 0.09 g) (p = 0.67) consumed only a low amount of normal food, indicating that they were close to satiety with respect to low-salience food.

Therefore, chronic exposure to an aversive social environment leads to: (1) decreased gustatory reward interest/learning where reinforcement rate can be maximized by discriminative learning under low-effort conditions; (2) decreased gustatory reward-to-effort valuation under conditions where reinforcement rate can be maximized by expending effort, but required effort increases progressively.

**Reduced DRLM or REV behaviours in chronic social stress mice co-occur with bi-directional activity changes in BA-NAc neurons.** To investigate whether changes in BA-NAc neuron

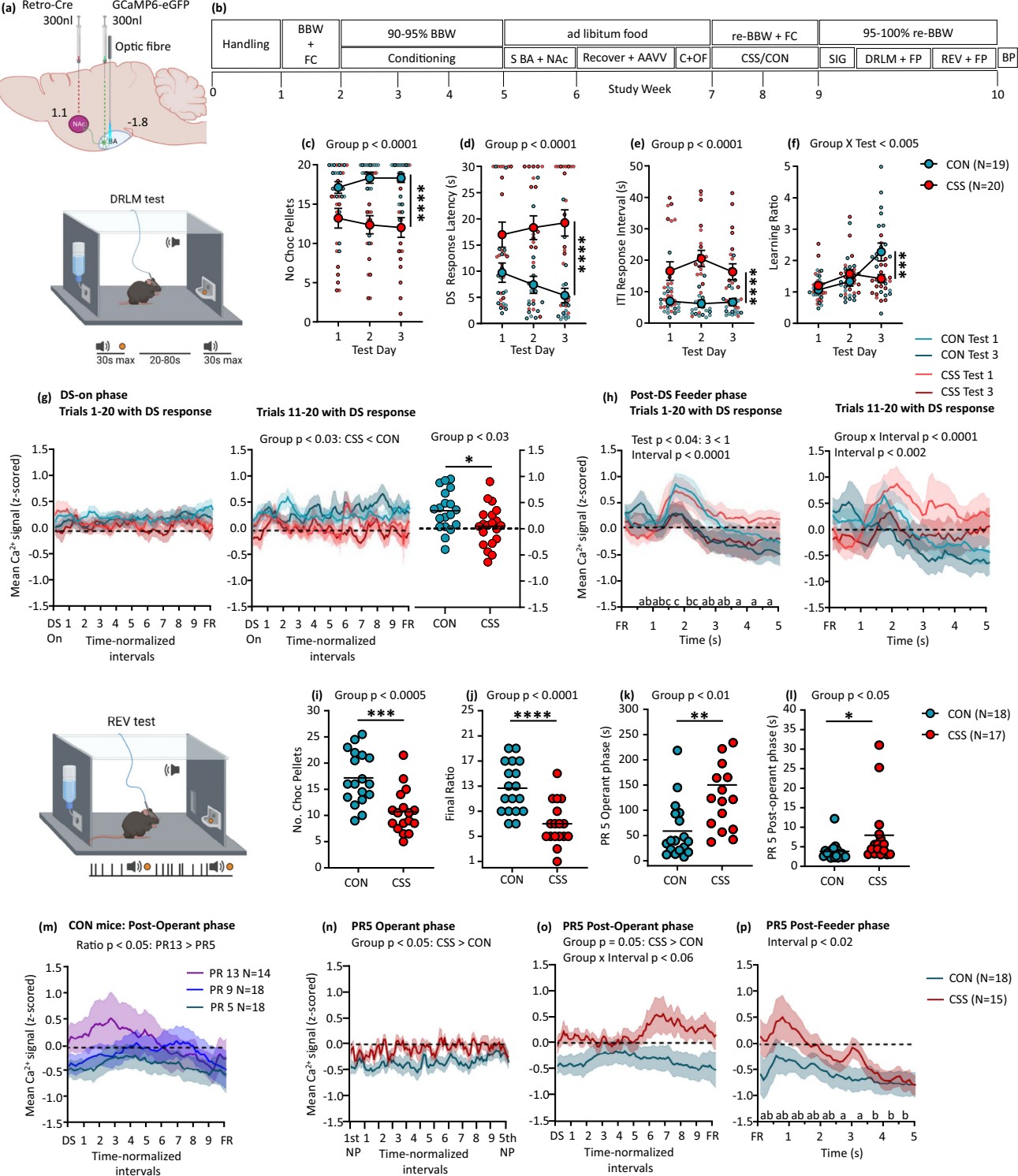

activity accompany the CSS-induced reductions in DRLM and REV, calcium-sensor fibre photometry was integrated into the CSS-reward behaviour model. Mice were injected unilaterally in the intermediate BA (bregma −1.8 mm) with a FLEX Cre-On vector encoding the calcium indicator *GCaMP6* (rAAV-FLEX-GCaMP6), followed by implantation of a fibre-optic probe, and injected in the ipsilateral NAc (bregma 1.1 mm, primarily core) with retrograde rAAV-retro-Cre-mCherry; this led to the high and specific expression of GCaMP6 in BA-NAc neurons (Fig. 2a, supplementary Fig. 1a–c). Coordinates were selected

based on the evidence that, at this bregma level, the BA is relatively large and contains a relatively high density of both monovalent reward- or aversion-responsive neurons[12,15,17]. Mice underwent conditioning for DRLM and REV tests, followed by stereotactic surgery (Fig. 2b). Following recovery, mice were given a session at the final conditioning stage for each behavioural test with the fibre photometry patch cord attached, to adjust them to making operant and feeder responses under fibre photometry conditions. Mice then underwent the CSS or CON procedure.

**Fig. 2 Effects of chronic social stress on BA-NAc neural activity during reward-directed behaviour. a** Schematic showing unilateral injection site of FLEX Cre-On GCaMP6 AAV vector in BA, fibre-optic probe implantation dorsal to the injection site, and ipsilateral injection of retrograde Cre AAV vector in NAc (relative to bregma). **b** Experimental design. BBW + FC: measurement of baseline body weight and food consumption; 90–95% BBW: conditioning under food restriction that reduced BW to 90–95% BBW; S BA + NAc: stereotactic surgery; Recovery + AAVV: recovery from surgery and AAV vector expression; C + OF: Conditioning sessions with patch cord attached to optic fibre; CSS/CON: CSS protocol or control handling; re-BBW + FC: BW and food consumption under ad libitum feeding on days 1–12 of CSS/CON provided re-baseline values; 95–100% re-BBW: mice were mildly food restricted to be tested at 95–100% re-BBW; SIG: fibre photometry signal test; BP: brains were perfused for histology. **c–h** DRLM test with fibre photometry. Mice received 3 daily tests of 31 trials each and trials 1–20 per test were used for data analysis. Data are shown as group mean ± SEM and per mouse scores. Behaviour: **c** Number of chocolate pellets obtained, i.e. DS trials with a response. **d** Median DS response latency. **e** Median ITI response interval. **f** Median learning ratio. Statistical analysis was conducted using 2-way mixed-model ANOVA. BA-NAc neural activity ($z$-scored $Ca^{2+}$ signal) relative to baseline, in tests 1 and 3: **g** For trials with a DS response, $Ca^{2+}$ activity during the DS-on phase (time-normalized and subdivided into 10 equal intervals) from DS-onset to feeder response for trials 1–20 (left) and trials 11–20 (centre), and non-time normalized mean values (right). **h** For trials with a DS response, $Ca^{2+}$ activity during the post-DS feeder phase which started with a feeder response and had a duration of 5 s (subdivided into 0.5 s intervals), for trials 1–20 (left) and trials 11–20 (right). Statistical analysis was conducted using 3-way mixed-model ANOVA. Test days and intervals indicated by different letters were significantly different in Tukey's multiple comparisons tests: e.g. a vs. b, a vs. c, b vs. c, $p < 0.05$ or lower. **i–p** REV test with fibre photometry. Mice received 3 daily tests and the mean of data for tests 2 and 3 are given and were used for statistical analysis. Data are shown as individual values or group mean ± SEM. Behaviour: **i** Number of chocolate pellets earned. **j** Final ratio attained. **k** At progressive ratio 5 (PR 5), operant phase duration i.e. time from 1st until 5th nosepoke. **l** At PR 5, post-operant phase duration i.e. time from DS onset until feeder response. Statistical analysis was conducted using $t$-tests. BA-NAc neuron population activity ($z$-scored $Ca^{2+}$ signal) relative to baseline: **m** In CON mice specifically, comparison of $Ca^{2+}$ activity during the post-operant phase—time from DS onset to feeder response (time-normalized and subdivided into 10 equal intervals)—at PRs 5, 9 and 13. **n** $Ca^{2+}$ activity during the operant phase at PR 5. **o** $Ca^{2+}$ activity during the post-operant phase at PR 5. **p** $Ca^{2+}$ activity during the post-feeder phase at PR 5. Statistical analysis was conducted using 2-way mixed-model ANOVA. Test days and intervals indicated by different letters were significantly different in Tukey's multiple comparisons test: e.g. a vs. b, a vs. c, b vs. c, $p < 0.05$ or lower. Images a, DRLM test and REV test were created with BioRender.com.

After a 1-day interval, mice were studied in the DRLM test on 3 consecutive days. The behavioural findings were similar to those obtained in the previous experiment, indicating that the integration of fibre photometry did not have a confounding effect (Fig. 2c–f, supplementary Tables S1, S2). Concerning BA-NAc neural pathway $Ca^{2+}$ activity in these behaving mice, the data are presented for DRLM tests 1 and 3 and are specific to those trials in which mice made a DS feeder response and therefore obtained a pellet. Trial-specific mean $Ca^{2+}$ activity during the 10 s prior to DS onset provided the measure of baseline activity (mean $z$-scored $Ca^{2+}$ signal = 0; Fig. 2g, h). The interval between DS onset and feeder response, referred to as the DS-on phase, was time normalized and divided into 10 equal intervals. Across trials 1–20 with a DS response, DS-on phase BA-NAc neural activity did not differ between groups or tests (Fig. 2g, left). To increase sensitivity to detecting effects attributable to within-session learning, analysis was restricted to trials 11–20 with a DS response: now, BA-NAc neural activity was decreased in CSS compared with CON mice (Fig. 2g, centre); this CSS effect also pertained when non-time-normalized mean $Ca^{2+}$ activity scores were analysed (Fig. 2g, right). In trials 1–10 with a DS feeder response, there was no CSS effect (supplementary Fig. 2a). The within-session divergence of BA-NAc neural $Ca^{2+}$ activity in CSS and CON mice can also be visualized when the data are presented for specific trials: supplementary Fig. 2b gives the $Ca^{2+}$ activity, for those mice which made a DS feeder response, for trials 1, 5, 10, 15 and 20, separately. The CSS effect on BA-NAc neural $Ca^{2+}$ activity was specific to trials in which mice made a DS feeder response, with there being no significant difference between CSS and CON mice in trials without a DS feeder response (supplementary Fig. 2c). After a DS feeder response, the trial progressed into the post-DS feeder phase which had a duration of 5 s (10 × 0.5 s intervals). As for the DS-on phase, trial-specific mean $Ca^{2+}$ activity during the 10 s prior to DS onset provided the measure of baseline activity. Considering post-DS feeder phases in trials 1–20, BA-NAc neural activity decreased from test 1 to 3 in both groups; activity was relatively high at time 1.5 s, coincident with pellet retrieval/consumption, and then declined to below baseline in CON mice and remained at baseline in CSS mice (Fig. 2h, left). In trials 11–20 specifically, BA-NAc

neural activity decreased from test 1 to 3, and whilst activity declined across time in CON mice it remained stable and therefore relatively high in CSS mice (Fig. 2h, right). In trials 1–10, BA-NAc neural activity declined from test 1 to test 3, and similarly in CSS and CON mice (Supplementary Fig. 2a). That post-DS feeder phase BA-NAc neural activity was related to reward retrieval/consumption is demonstrated by comparison with feeder responses during ITIs, when activity remained at or below baseline (Supplementary Fig. 2d). Furthermore, in trials 11–20, ITI feeder phase activity was lower in CSS than CON mice, suggesting that absence of reward was less expected. Therefore, CSS effects on BA-NAc neural activity emerged within DRLM sessions: during the DS, BA-NAc neural activity was lower in CSS than CON mice, and this co-occurred with a longer DS response latency that did not change across tests in CSS mice. During the post-DS feeder phase, BA-NAc activity declined by test 3, and whilst it declined below baseline in CON mice it remained at baseline in CSS mice. Overall, these findings suggest that CSS leads to reduced activation of reward-responsive BA-NAc neurons by the tone DS—this reduction during discriminative reward processing might causally contribute to the CSS-induced decrease in reward interest/learning in the DRLM test.

Mice proceeded on the next day to the REV test: 3 tests were conducted and for each subject the means of data obtained in tests 2 and 3 were analysed. Behavioural findings were similar to those obtained in the previous experiment (Fig. 2i–l, Supplementary Fig. 3a). Trial-specific mean BA-NAc activity during 10 s prior to the onset of operant responding constituted baseline activity, and was used for each phase of analysis. Three successive phases were analysed: operant phase, comprising 10 time-normalized intervals from the first to the last response per trial; post-operant phase, 10 time-normalized intervals from the onset of 1-s tone DS to feeder response; post-feeder phase, from directly after feeder response for a duration of 5 s (10 × 0.5 s intervals). Firstly, to investigate BA-NAc neural activity under conditions of progressively increasing effort in CON mice specifically, activity in the post-operant phase was compared at PR 5, 9 and 13: interestingly, neural activity increased with effort; furthermore, in absolute terms it was below baseline at PR 5, at baseline at PR 9 and above baseline at PR 13 (Fig. 2m). All 18 CON mice and

15 of 17 CSS mice reached at least progressive ratio 5 (PR 5) (Fig. 2j) and this ratio was used to investigate CSS effects: The operant phase was longer in CSS than CON mice (Fig. 2k) and operant phase BA-NAc neural activity was higher in CSS than CON mice (Fig. 2n). The post-operant phase was slightly longer in CSS than CON mice (Fig. 2l) and during this phase BA-NAc neural activity was higher in CSS than CON mice (Fig. 2o). (It is noteworthy that mean BA-NAc neural activity during the post-operant phase was also higher in CSS than CON mice at PR 9 and PR 13 (supplementary Fig. 3b) but due to the low sample size for CSS mice at these higher PRs statistical analysis was not conducted.) Post-feeder neural activity was similar in CSS and CON mice, with both groups displaying the highest activity at the time of pellet retrieval/consumption followed by a decline to below baseline (Fig. 2p). Therefore, in CON mice, at PR 5 the activity of BA-NAc neurons was sub-baseline across the 3 test phases, indicating that at least some neurons were inhibited during engagement in increasingly effortful operant behaviour. Increases in progressive ratio and therefore effort led to higher BA-NAc neural activity in the post-operant phase, to a level above baseline by PR 13, indicating that at least some neurons were activated in response to increased effort. Comparing CSS with CON mice at PR 5, BA-NAc neural activity during operant and post-operant phases was higher in CSS than CON mice, suggesting that CSS increases sensitivity of some neurons to effort —intriguingly, the effort-related activity of CSS mice at PR 5 was equivalent to that of CON mice at PR 13. Such an increase in the activity of apparently aversion-responsive BA-NAc neurons during effort processing might contribute causally to the CSS-induced decrease in reward-to-effort valuation in the REV test.

When comparing the two CSS experiments, there were some differences in the absolute behavioural scores in the case of CON mice and CSS mice, which are likely attributable to the various effects of the procedures specific to the fibre photometry experiment (e.g. optic fibre implantation, patch cord attachment, modification of reward dispenser). However, the major, consistent finding from the two experiments is that, even under these different test conditions, the model of CSS-induced reductions in DRLM and REV is reproducible and robust.

**Chronic social stress has a limited effect on population-level transcriptome expression of BA-NAc neurons.** Having identified that CSS leads to decreased BA-NAc neuron activity during a reward DS (DRLM test) and increased BA-NAc neuron activity during effortful behaviour (REV test), we next investigated whether CSS induces changes in the transcriptome of BA-NAc neurons, for example in genes encoding proteins involved in glutamate signalling (Fig. 3a). Mice were injected bilaterally in NAc (bregma 1.1 mm) with retrograde neuronal tracer cholera toxin subunit β conjugated with Alexa Fluor-555 (CTB-555) (Fig. 3b). They then underwent CSS or CON, and on the following day, perfusion with phosphate-buffered saline (PBS) to rinse the blood from the brains, which were then fresh frozen. Coronal 10 µm sections including intermediate BA were collected (Fig. 3c) and mounted onto PET membrane slides and dehydrated-fixed. Laser capture microdissection was used to collect neuron soma-size areas of BA tissue that were CTB-555[+] (Fig. 3d, e). For each mouse, 500 such putative BA-NAc neuron samples were collected and lysed using RLT buffer. Lysates were pooled for RNA extraction and low-input RNA sequencing libraries were prepared with the SMART-seq v4 FLX protocol and sequenced with the Illumina NovaSeq platform. After filtering out genes with low expression, a median of >13,000 genes was detected per mouse. To determine whether samples comprised primarily BA-NAc neuron somata, expression levels of brain cell type-specific marker genes were compared (Fig. 3f): expression levels of neuron gene Snap25 (synaptosomal associated protein 25 gene) and glutamate-neuron gene Slc17a7 (vesicular glutamate transporter 1 gene) were relatively high, whereas expression levels of marker genes for all other cell types were low. The genes Ppp1r1b (protein phosphatase 1, regulatory inhibitor subunit 1b) and Rspo2 (R-spondin-2), proposed as marker genes for BA reward and aversion neurons, respectively[15], were both expressed (Fig. 3f). Principal component analysis (PCA) identified the absence of clear separation of BA-NAc neuron transcriptome expression in CSS and CON mice. Differential gene expression analysis (DGEA) was conducted at thresholds of absolute log2-fold change (FC) > 0.5 and nominal $p < 0.001$: this identified 2 down-regulated and 64 up-regulated genes in CSS compared with CON mice (Fig. 3g). Functional enrichment analysis (FEA) with mouse-specific KEGG pathways identified that the genes up-regulated in CSS mice were enriched (false discovery rate (FDR) $p < 0.05$) in the gene sets, Rap1 signalling pathway (Pdgfrb, Pard3, Id1, Adcy5), Axon guidance (Sema3c, Pard3, Plxnb2, Myl9, Robo1) and MAPK signalling pathway (Pdgfrb, Gja1, Adcy5). Individual genes of interest that were up-regulated in BA-NAc neurons from CSS mice included: Gata2, encoding transcription factor GATA-binding factor 2, increased expression of which resulted in impaired dendritic outgrowth and spine formation in adult mice;[27] Timp3, encoding tissue inhibitor of metalloproteinase 3 and knockout of which led to learning deficits;[28] and Plxnb2, encoding the Plexin-B2 receptor which contributes to aversion-induced neuronal structural plasticity by increasing dendrite ramifications and modulating synaptic density[29].

**Chronic inhibition of BA-NAc neural pathway replicates the reduced DRLM of CSS mice.** CSS reduces DRLM coincident with decreased BA-NAc neural activity, effects which might be mediated by BA-NAc reward-responsive neurons. In the next experiment, we investigated whether specific and chronic inhibition of BA-NAc neurons causes a similar behavioural effect in the DRLM test specifically. Mice were injected bilaterally in BA (bregma −1.8 mm) with an AAV vector encoding tetanus toxin light chain (TeTxLC) in a Cre-dependent manner (rAAV-dlox-TeTxLC-eGFP). In glutamate neurons among others, TeTxLC cleaves the SNARE protein, vesicle-associated membrane protein 2 (VAMP2, synaptobrevin), and thereby reduces vesicular docking/fusion steps of pre-synaptic neurotransmitter exocytosis[30,31]. To induce TeTxLC expression in BA-NAc neurons specifically, mice were injected bilaterally in NAc (bregma 1.1 mm, primarily core; supplementary Fig. 5a–c) with rAAV-retro-Cre-mCherry. Pilot experiments were conducted to establish an AAV vector titre that resulted in the decreased intensity of VAMP2 immunofluorescence signal in the BA, whilst also ensuring that there was no effect on basal locomotor behaviour (Supplementary Fig. 4a–g). In the main experiment, two control groups were deployed: mice injected in BA with the vehicle and in NAc (primarily core; supplementary Fig. 5c) with rAAV-retro-Cre-mCherry, and mice injected in BA with rAAV-dlox-TeTxLC-eGFP and in NAc with rAAV-retro-mCherry (in both control groups, mice were also injected in BA with fluorescent tracer for injection site validation) (Fig. 4a, j). As for CSS/CON experiments, mice were first conditioned in preparation for DRLM and REV testing, and they then underwent stereotactic surgery for rAAV vector injection (Fig. 4b). After 15 days, corresponding to the duration of CSS, mice were studied in the DRLM test on 4 consecutive days (Supplementary Table S1). The number of trials with a DS feeder response per test was similar across the 3 groups with an overall mean of 29.8 ± 2.0 (Supplementary Table S2), and data analysis was conducted for trials 1–20 per test. There was no

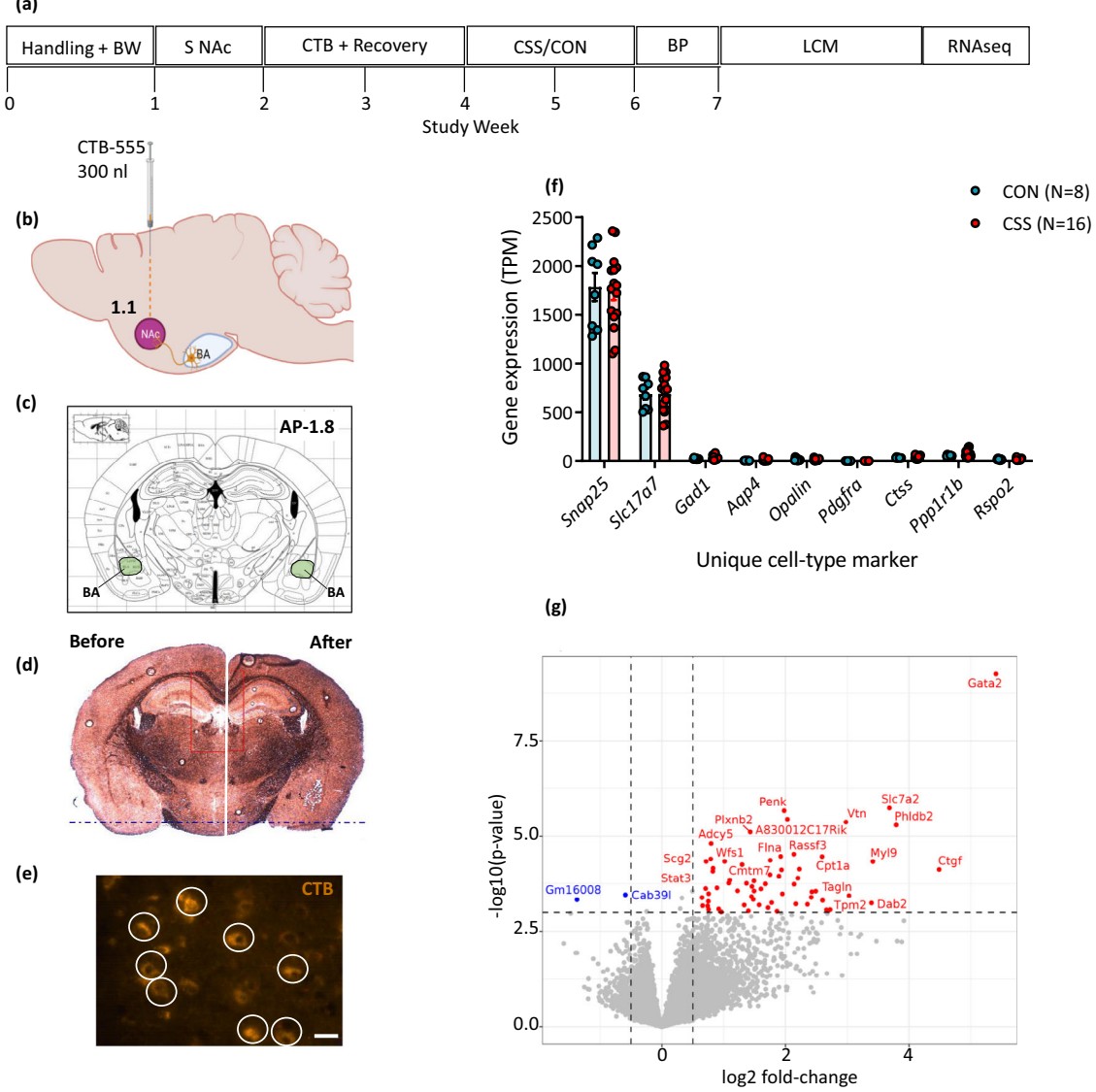

**Fig. 3 Effects of CSS on population-level transcriptome expression of BA-NAc neurons. a** Experimental design. Handling + BW: daily handling and measurement of body weight; S NAc: stereotactic surgery; CTB + Recovery: CTB transport and recovery from surgery; CSS/CON: CSS protocol or control handling; BP: brains were PBS perfused; LCM: collection of BA-NAc CTB⁺-labelled tissue using laser capture microdissection; RNA-seq: RNA sequencing and differential gene expression analysis. **b** Schematic showing bilateral injection site of retrograde CTB Alexa Fluor-555 in NAc. **c** Coronal image from mouse brain atlas[62] at bregma level −1.8 mm with BA highlighted. **d** Representative coronal image (×5) from brain of a CSS mouse at bregma −1.8 mm before (left) and after (right) collection of CTB-555⁺ tissue samples using LCM. **e** Representative coronal image (×40) from a CSS mouse BA at bregma −1.8 mm: white circles indicate areas of CTB-555⁺ tissue demarcated for LCM. Scale bar = 200 μm. **f** Expression levels (transcript per million) of cell type-specific marker genes: *Snap25* neuron, *Slc17a7* glutamate neuron, *Gad1* GABA neuron, *Aqp4*, astrocyte, *Opalin* myelinating oligodendrocyte, *Pdgfra* oligodendrocyte progenitor cell, *Ctss* microglia. **g** Volcano plot for differential gene expression in CSS compared with CON mice: significantly up-regulated genes are shown in red and significantly down-regulated genes in blue. Image **b** was created with BioRender.com. Image **c** was used with permission of Elsevier, from The Mouse Brain Atlas, G. Paxinos & K.B.J. Franklin, 2nd edition, 2001; permission conveyed through Copyright Clearance Center, Inc.

significant effect of BA-NAc neuron TeTxLC on the number of DS responses and therefore rewards obtained (Fig. 4c), DS response latencies (Fig. 4d), or ITI response intervals (Fig. 4e). Nonetheless, there was an effect of BA-NAc neuron TeTxLC on the discriminative learning ratio (Fig. 4f): whilst the ratio was close to 1 in all 3 groups in test 1 and then increased progressively across tests in most control mice—primarily due to an increase in ITI response interval—it remained close to 1 in most TeTxLC mice. In DRLM test 4, mice were provided with a pellet of normal food and mice in each group consumed a low and similar amount indicating that they were all close to satiety with respect to low-salience food (Supplementary Fig. 4h). Mice were then studied in

the REV test on 2 consecutive days and the data for test 2 were used to assess for BA-NAc neuron TeTxLC effects. Compared with control mice, TeTxLC mice tended to complete fewer responses (Supplementary Fig. 4i), they earned fewer rewards (Fig. 4g), and tended to attain a lower final progressive ratio (Fig. 4h); there was no TeTxLC effect on pellet retrieval latency or post-reinforcement pause (Supplementary Fig. 4j, k).

Therefore, expressing TeTxLC in BA-NAc neurons with the aim of chronically inhibiting their glutamate release replicates some specific effects of CSS on reward-directed behaviour. Namely, in the DRLM test, TeTxLC mice displayed reduced reward learning, similar to that of CSS. In the REV test, on most

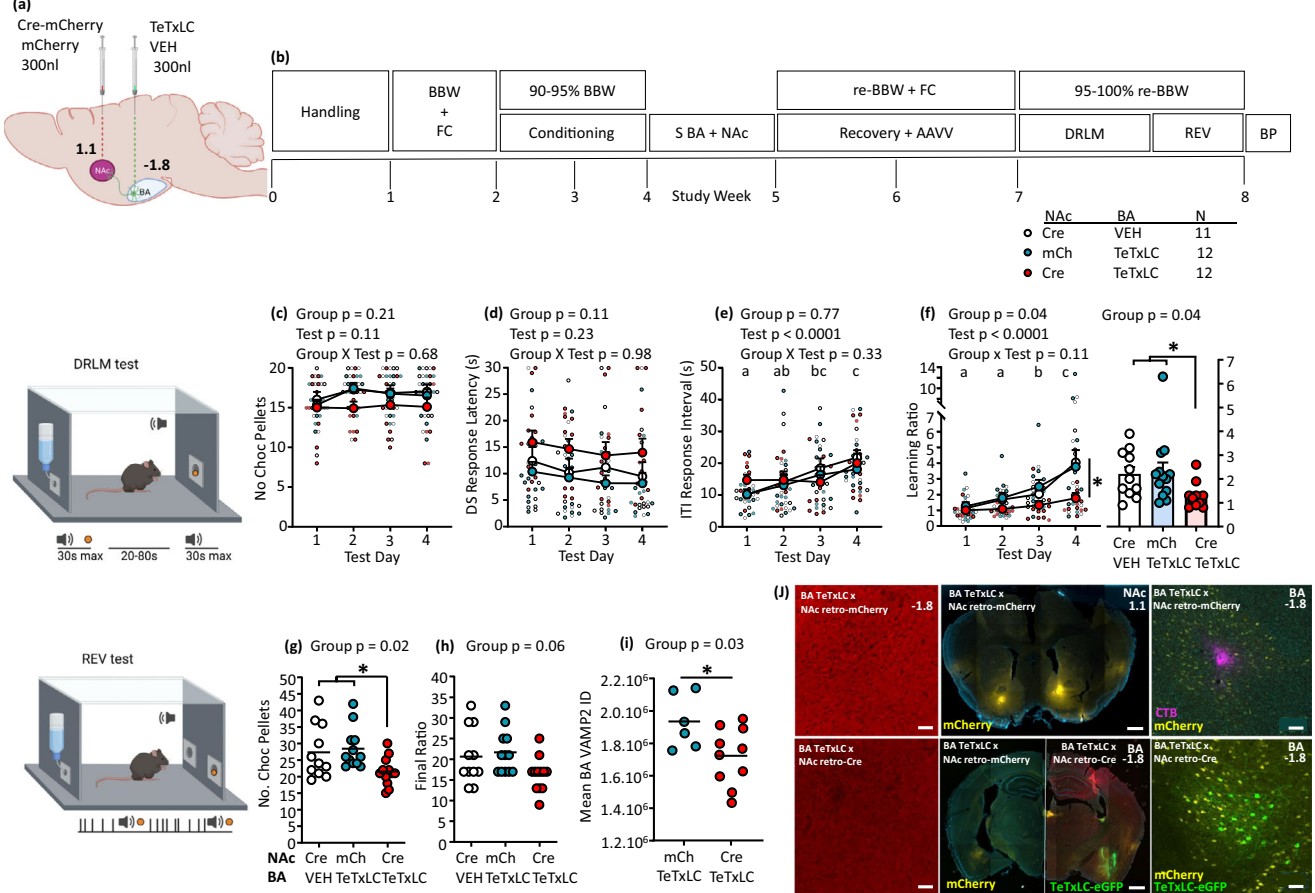

**Fig. 4 Effects of tetanus-toxin-light-chain inhibition of BA-NAc neurons on reward-directed behaviour. a** Schematic showing bilateral injection site of Cre-dependent TeTxLC AAV vector or vehicle in the BA and of retrograde-Cre AAV vector or retrograde-mCherry AAV vector in the NAc (relative to bregma). **b** Experimental design. BBW + FC: measurement of baseline body weight and food consumption; 90–95% BBW: conditioning under food restriction to reduce BW to 90–95% BBW; S NAc+BA: stereotactic surgery; Recovery+AAVV: recovery from surgery and AAV vector expression; re-BBW + FC: BW and food consumption under ad libitum feeding on days 5–13 post-surgery provided re-baseline values; 95–100% re-BBW: mice were mildly food restricted to be tested at 95–100% re-BBW; BP: brains were perfused for histology. **c–f** DRLM tests on 4 days of 40 trials each and trials 1–20 per test were used for data analysis. Data are shown as group mean + SEM and per mouse scores: **c** Number of chocolate pellets obtained i.e. DS trials with a response. **d**. Median DS response latency. **e**. Median ITI response interval. **f** Median learning ratio, across tests (left) and individual mean scores for all tests (right). **g** and **h** REV tests on 2 days and data for test 2 are shown as individual scores and group means: **g** Number of chocolate pellets earned. **h** Final ratio attained. **I**, **j** Histology and immunofluorescence: **i** Integrated density of VAMP2 immunofluorescent signal in the entire BA at bregma AP −1.8 mm in BA TeTxLC × NAc retro-Cre mice versus BA TeTxLC × NAc retro-mCherry mice. Data are individual and overall group means. **j** Representative confocal micrographs for: Left upper: BA with VAMP2 immunosignal (×20) in a control mouse. Left lower: BA with VAMP2 immunosignal (×20) in a TeTxLC mouse. Scale bar = 50 μm. Middle upper: NAc injection site (×5). Middle lower: Left, BA with mCherry in a control mouse (×5). Scale bar = 500 μm; Right, BA with eGFP in a TeTxLC mouse (×5). Right upper: BA with mCherry and CTB-555 in a control mouse (×20). Right lower: BA with eGFP in a TeTxLC mouse (×20). Scale bar = 50 μm. Statistical analysis for the DRLM test was conducted using 2-way mixed-model ANOVA and for the REV test with 1-way ANOVA. In the case of a significant group effect, planned contrasts were conducted: in the absence of a difference between the 2 control groups, they were combined, and a control vs. TeTxLC contrast was conducted. Statistical analysis for VAMP2 integrated density was conducted using unpaired t test. Test days indicated by different letters were significantly different in Tukey's multiple comparisons test: e.g. a vs. b, a vs. c, b vs. c, p < 0.05 or lower. Images a, DRLM test and REV test were created with BioRender.com.

measures, there was a trend-level effect of TeTxLC in BA-NAc neurons, whilst TeTxLC mice did earn fewer rewards. Integrating these findings with those obtained in the CSS-reward test-fiber photometry experiment, it can be hypothesized that chronic BA-NAc neuron inhibition leads to decreased reward learning due primarily to decreased activity in reward-sensitive neurons specifically.

To obtain evidence for the TeTxLC-mediated reduction of VAMP2, two approaches were used. Representative images showing the coronal distribution of TeTxLC-targeted BA-NAc neurons in the BA are given in supplementary Fig. 5. Firstly, the intensity of VAMP2 immunofluorescence signal was compared in

TeTxLC and control mice. Confocal images of VAMP2 immunostaining demonstrated decreased VAMP2 signal in the BA-NAc neuron-rich region in TeTxLC mice compared with control mice (Fig. 4i, j; Supplementary Fig. 4f, g). Second, to obtain a TeTxLC readout specific to BA-NAc neurons, the transcriptome of BA-NAc neurons was investigated in a separate cohort of mice (Supplementary Fig. 6a–c). Samples were collected as described above for the CSS BA-NAc neuron transcriptome experiment, except that now EGFP[+] tissue was collected in the case of TeTxLC mice and mCherry[+] tissue for control mice (supplementary Fig. 6d, e), to obtain a total of 500 neuron samples per mouse, and QIAzol was used for lysis. RNA

extraction and low-input RNA library preparation and sequencing were conducted. After filtering out genes with low expression, a median of >13,000 genes was detected per mouse. Expression levels of *Snap25* and *Slc17a7* were relatively high; this was the case in both TeTxLC mice and control mice, although *Slc17a7* expression was reduced in the former (Supplementary Fig. 6f). The astrocyte marker gene *Aqp4* and microglia marker gene *Ctss* were expressed more highly in TeTxLC than control mice. These findings confirm that most tissue sampled was BA-NAc glutamate neurons somata, and that co-collection of astrocyte and microglia tissue was increased in TeTxLC mice. Supplementary Fig. 6i presents a confocal image of a coronal brain section including BA from a mouse injected in NAc with the retrograde tracer cholera toxin B-fluorophore 555 in which immunostaining for the microglia marker IBA1 was conducted; even in the absence of TeTxLC, the close proximity of microglial processes to glutamate neuron somata is apparent. PCA identified a clear separation between TeTxLC and control mice. DGEA (absolute log2 FC > 0.5, nominal $p < 0.001$) identified 23 downregulated and 249 up-regulated genes in TeTxLC mice (supplementary Fig. 6g, h). FEA with mouse-specific KEGG pathways identified that the genes up-regulated in TeTxLC mice were enriched (FDR $p < 0.05$) in the gene sets Lysosome, Cellular senescence, Apoptosis (supplementary Fig. 7), Chemokine signalling pathway and Phagosome, among others. Taken together, the findings are consistent with the following sequence of events in BA-NAc neurons: TeTxLC-induced VAMP2 cleavage, glutamate accumulation, neuronal senescence and apoptotic processes, and paracrine activation of glial cells.

**Chronic activation of BA-NAc neurons replicates the reduced REV of CSS mice.** CSS reduces REV coincident with increased BA-NAc neural activity, effects which might be mediated by BA-NAc aversion-responsive neurons. CSS also induced increased expression of genes encoding proteins involved in secondary signalling pathways in BA-NAc neurons. In the final experiment, we investigated whether specific and chronic activation of BA-NAc neurons causes a similar behavioural effect in the REV test. Mice were injected bilaterally in the BA (bregma −1.8 mm) with a Cre-dependent vector encoding the excitatory DREADD *rM3D(Gs)* (ssAAV-dlox-rM3D(Gs)-eGFP). To induce rM3D(Gs) expression in BA-NAc neurons, mice were injected bilaterally in the NAc (bregma 1.1 mm, primarily core; Supplementary Fig. 5f) with rAAV-retro-Cre-mCherry (Fig. 5a). Pilot experiments were conducted to establish an appropriate AAV vector titre that was efficacious, whilst also ensuring that there was no effect on basal locomotor behaviour (Supplementary Fig. 8a–e). Because both clozapine (CLZ) and its precursor CLZ-N-oxide (CNO) were used as DREADDs actuators, pilot studies were conducted to establish doses of both that were without effects on basal locomotor behaviour (Supplementary Fig. 8f–j). To determine whether the indicated CLZ dose (0.3 mg/kg) was efficacious in activation of rM3D(Gs)-expressing BA-NAc neurons, pilot rM3D(Gs) mice and control mice were injected with CLZ and brains processed for immunostaining of c-Fos, the protein encoded by the immediate early gene *Fos*. Mice expressing rM3D(Gs) in BA-NAC neurons demonstrated a marked increase in c-Fos$^+$ BA-NAc neurons compared with controls. Interestingly, the number of c-Fos$^+$ BA cells was substantially higher than that of c-Fos$^+$ BA-NAc neurons, suggesting that activation of the latter also impacted on neighbouring neurons (Supplementary Fig. 8k–m).

In the main experiment (Fig. 5b) mice received: (1) continuous, low-level activation of BA-NAc neurons, as hypothesized to occur during distal sensory exposure phases of CSS, achieved by administering CNO in the drinking water (0.2 mg/kg/day); (2) acute activation of BA-NAc neurons, as hypothesized to occur during the proximal attack phases of CSS, achieved by injecting CLZ once per day (0.3 mg/kg). Two control groups were required: mice injected in BA and NAc with the same viral vectors as rM3D(Gs) × CNO/CLZ mice but that received normal drinking water and vehicle injections only, and mice injected in BA with rM3D(Gs) and in NAc with rAAV-retro-mCherry only, and that received CNO in water and CLZ injections. After 15 days, corresponding to the duration of CSS, behavioural testing began, with CNO and CLZ (15 min pre-test) administration continuing on test days (supplementary Table S1). Four daily DRLM tests were conducted: mice in the three groups made a similar number of DS responses per test with an overall mean of $27.7 ± 2.9$ (Supplementary Table S2). Statistical analysis was conducted with the data for trials 1–20 per test. The rM3D(Gs)×CNO/CLZ mice made fewer DS responses and therefore obtained fewer rewards than control mice (Fig. 5c). The DS response latencies tended to be longer in rM3D(Gs)×CNO/CLZ mice than control mice (Fig. 5d). The intervals between ITI responses were longer in rM3D(Gs)×CNO/CLZ mice than No rM3D(Gs)×CNO/CLZ mice (Fig. 5e). Because rM3D(Gs)×CNO/CLZ mice were not only relatively slow in responding to the DS but also in ITIs, there was no effect of rM3D(Gs)×CNO/CLZ on the learning ratio, and all groups displayed an increase in the learning ratio across tests (Fig. 5f). In test 4, a pellet of normal food was provided and mice in each group consumed a low amount indicating that they were all close to satiety with respect to low-salience food (Supplementary Fig. 9b). Mice were then studied in the REV test on 3 consecutive days and the mean scores of tests 2 and 3 were used for statistical analysis. The rM3D(Gs)×CNO/CLZ mice tended to complete fewer operant responses than control mice (supplementary Fig. 9c); they earned fewer rewards (Fig. 5g) and attained a lower final progressive ratio (Fig. 5h). There was no effect of rM3D(Gs)×CNO/CLZ on pellet retrieval latency or post-reinforcement pause (supplementary Fig. 9d, e). Following completion of behavioural testing, some rM3D(Gs)×CNO/CLZ mice and No-rM3D(Gs)×CNO/CLZ mice continued to receive CNO for 7 days and then received a final CLZ injection. In these mice, to determine whether the DREADDs system was still functional after chronic actuation, the number of BA-NAc c-Fos$^+$ neurons were compared: this was markedly higher in the rM3D(Gs)×CNO/CLZ mice (Fig. 5i, j). Representative images showing the coronal distribution of DREADDs-targeted BA-NAc neurons in the BA are given in Supplementary Fig. 5d–f; the distribution and density are similar to those obtained in the TeTxLC BA-NAc neuron inhibition experiment (Supplementary Fig. 5a–c).

Therefore, expressing excitatory DREADD in BA-NAc neurons with the aim of chronic activation replicates some specific effects of CSS on reward-directed behaviour. Namely, in the DRLM test, although measures of reward interest were reduced, there was no effect on reward learning, in contrast to CSS. In the REV test, both the number of rewards earned and the final ratio attained were decreased, similar to CSS. Integrating these findings with those obtained in the CSS-reward test-fibre photometry experiment, it can be hypothesized that chronic BA-NAc neuron activation leads to increased sensitivity to aversive effort due primarily to increased activity in aversion-sensitive neurons specifically.

## Discussion
Whilst there is unequivocal evidence that chronic experience of aversion ("stress") is a major aetiological factor for reward psychopathologies[5,32], biological and clinical understanding of

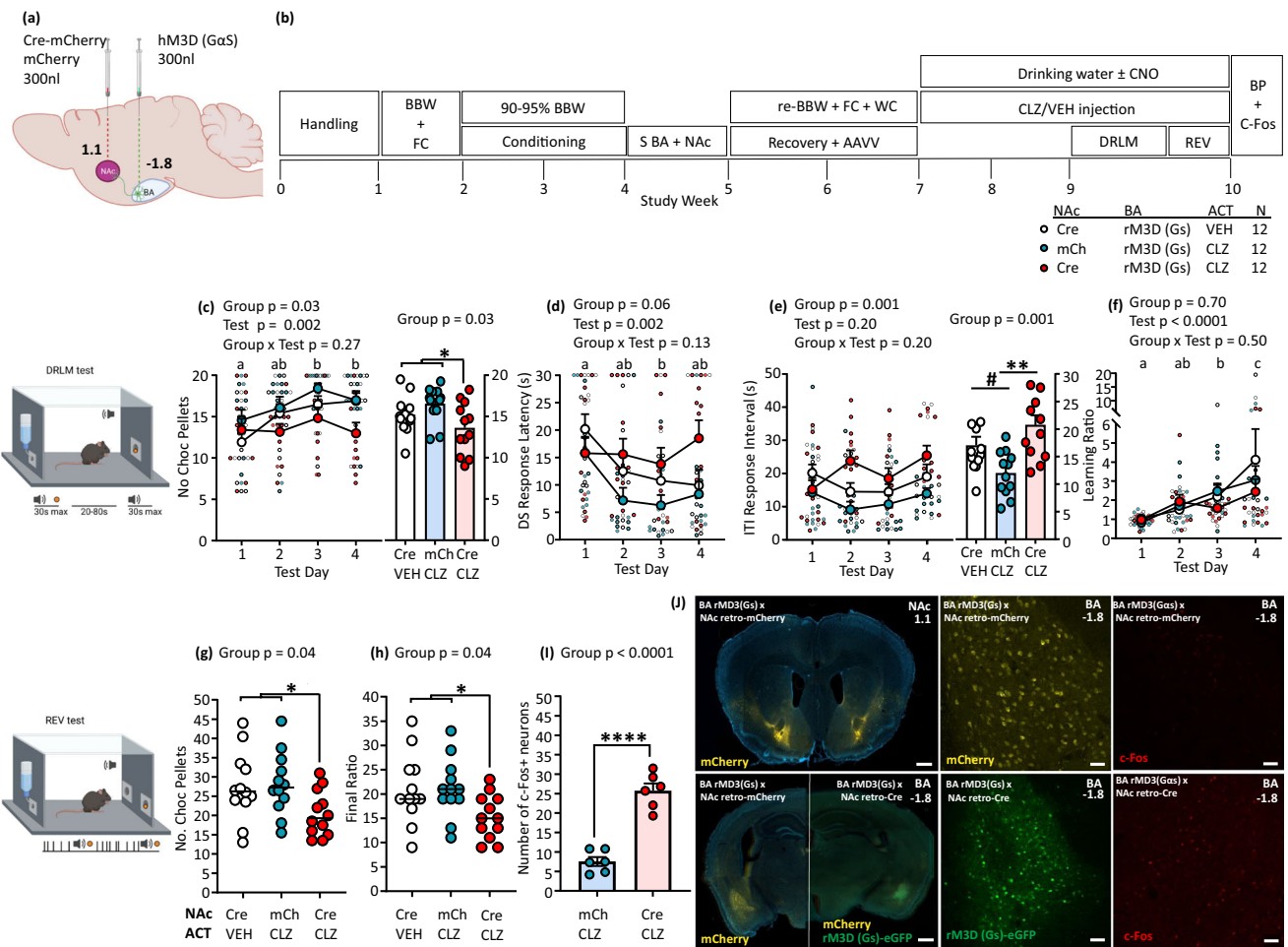

**Fig. 5 Effects of DREADDs activation of BA-NAc neurons on reward-directed behaviour. a** Schematic showing bilateral injection site of Cre-dependent rM3D(Gs) AAV vector in the BA and retrograde Cre AAV vector or retrograde mCherry AAV vector in the NAc (relative to bregma). **b** Experimental design. BBW + FC: measurement of baseline body weight and food consumption; 90–95% BBW: conditioning under food restriction to reduce BW to 90-95% BBW; S BA + NAc: stereotactic surgery; Recovery+AAVV: recovery from surgery and AAV vector expression; re-BBW + FC + WC: food and water consumption under ad libitum conditions on days 6–14 post-surgery provided re-baseline values and water intake for calculation of CNO administration; Drinking water +/- CNO: administration of CNO via drinking water depending on the group; CLZ/VEH injection: daily injection of CLZ or VEH depending on the group; BP: brains were perfused for histology. **c–f** DRLM tests on 4 days of 40 trials each and trials 1–20 per test were used for data analysis. Data are shown as group mean + S.E.M and per mouse scores: **c** Number of chocolate pellets obtained, i.e. DS trials with a response, across tests (left) and individual means for all tests (right). **d** Median DS response latency. **e** Median ITI response interval across tests (left) and individual means for all tests (right). **f** Median learning ratio. **g**, **h** REV tests on 3 days, and the mean of scores for tests 2 and 3 are shown and was used for statistical analysis. Data are shown as individual scores and group means. **g** Number of chocolate pellets earned. **h** Final ratio attained. **i**, **j** Histology and immunofluorescence: **i** Number of c-Fos+ neurons (bregma −1.8 mm) in rMD3(Gs) CNO/CLZ mice versus no rM3D(Gs) CNO/CLZ mice. **j**. Representative confocal micrographs for: Left upper: NAc injection site (×5). Left lower: Left, BA with mCherry in a control mouse (×5); Right, BA with eGFP in an experimental mouse (×5). Scale bar = 500 μm. Middle upper: BA with mCherry in a control mouse (×20). Middle lower: BA with eGFP in an experimental mouse (×20). Right upper: c-Fos signal in BA of a control mouse (×20). Right lower: c-Fos signal in BA of an experimental mouse. Scale bar = 50 μm. Statistical analysis for the DRLM test was conducted using 2-way mixed-model ANOVA and for the REV test with 1-way ANOVA. In the case of a significant group effect, planned contrasts were conducted: in the absence of a difference between the 2 control groups, they were combined, and a control vs experimental contrast was conducted. Statistical analysis for c-Fos+ neurons was conducted using an unpaired *t* test. Test days indicated by different letters were significantly different in Tukey's multiple comparisons test: e.g. a vs. b, a vs. c, b vs. c, *p* < 0.05 or lower. Images a, DRLM test and REV test were created with BioRender.com.

the mediating neural circuitry and pathophysiology is sparse. The amygdala, including BA long-range glutamate projection neurons, constitutes a major candidate in this regard: BA neurons receive glutamate-mediated inputs from lateral amygdala (LA) relating to unconditioned and conditioned aversion and reward[10,33]. Some BA glutamate neurons project to NAc (ventral striatum)[12,15,17,34], a pathway that integrates the BA directly with the basal ganglia-mesolimbic circuits of reward and aversion processing. These circuits involve bi-directional signalling between medium spiny neurons of the NAc and dopamine (DA) neurons of the ventral tegmental area (VTA)[35,36], as well as

between VTA and BA[37]. Characteristics of BA neurons vary along its rostral–caudal axis with respect to morphology, projection region, gene expression, and the emotional valence to which they respond[12,15,16,38,39]. The present mouse study focuses on the intermediate BA: this sub-region comprises anterior (magnocellular) and posterior (parvocellular) BA, has the highest absolute number of BA neurons projecting to the intermediate NAc core and shell, and a high proportion of these neurons are either reward or aversion responsive[12,15,16]. Evidence is provided for the importance of changes in these BA-NAc neurons in mediating the disruptive effects of chronic stress on distinct

aspects of reward processing. Based on histological findings, in the current study the majority of BA-NAc neurons studied projected to NAc core and the minority to NAc shell. As such, and similar to previous studies of BA-NAc neurons and reward-directed behaviour[12,15,17], it is not possible to attribute findings to BA projection to one particular NAc subregion; importantly, the core-shell distribution of the neurons studied was highly similar across groups within experiments and across experiments.

In young-adult male mice, the chronic social stressor used in this study leads to a generalized increase in the salience of aversive stimuli: for example, using mild foot shock as the US, CSS mice display increased Pavlovian (CS-US) conditioning and decreased operant (DS-response) active avoidance learning;[21,40] these behavioural processes are amygdala-dependent[1,41]. As demonstrated previously and here, CSS also leads to deficits in reward processing, using a sucrose US with mice in a close to satiated state[22,24,25,42,43]. These CSS effects on non-social behaviour are obtained without subgrouping mice according to whether or not they passively avoid CD-1 male mice subsequent to the CSS period, a procedure that is commonly used in other studies (e.g. refs. [44,45]). It is well-established that mice show marked spontaneous inter-individual differences (traits) in behaviour, including in passive avoidance; indeed, a recent study demonstrated that inter-individual differences in passive avoidance before chronic social stress positively predict inter-individual differences in passive avoidance after chronic social stress i.e. trait effects and chronic social stress effects are additive[46]. Rather than allocating mice to subgroups based on their passive avoidance trait/state, our approach is to investigate CSS effects in the entire sample of mice, so that trait differences are controlled for between the CSS and CON groups. As can be seen for behavioural data and fibre photometry data (Figs. 1 and 2), inter-individual variability is comparable between CSS and CON groups for nearly all measures, indicating that, for the reward processes investigated, there is no evidence for subgroups of CSS mice. Indeed, the same applies when the behavioural readout is Pavlovian aversion learning and memory (e.g. refs. [24,40]). As described in the "Results" section, all CSS mice displayed submissive behaviour during proximal attack sessions. Using this experimental design, sample sizes of 12 are sufficient to yield significant CSS effects without subgrouping.

The DRLM test is based on learning of associations between DS, appetitive behaviour and US; whilst it differs from CS-US learning, the attribution of emotional salience to a DS paired with a reward US is amygdala-dependent[15]. CSS mice display reduced DRLM relative to controls: whilst this is partly attributable to reduced interest and, therefore, exposure to the DS-response–US contingency, the marked reduction in the learning ratio indicates that DS-US learning per se is reduced in CSS mice. Recording of BA-NAc neural pathway $Ca^{2+}$ activity during the DRLM test demonstrated that CON mice undergo a within-session increase in activity during the DS-on phase; this co-occurred with and potentially contributed to, a between-session decrease in DS response latency and increase in learning ratio. That the increase in DS-on $Ca^{2+}$ activity constituted BA-NAc neuron reward prediction error (RPE)[47,48] is supported by decreased $Ca^{2+}$ activity in the post-DS feeder phase from session 1–3—a brief activity peak coincident with reward retrieval was followed by a decrease below baseline. This feeder-response profile of intermediate BA-NAc neural pathway $Ca^{2+}$ activity is similar to that described previously for unmanipulated (control) mice[49]. Compared with CON, DS-on phase $Ca^{2+}$ activity was low in CSS mice, coincident with and potentially causally contributing to their extended, intransigent DS response latency. Given the low DS-on phase $Ca^{2+}$ activity of CSS mice, according to RPE[48], their $Ca^{2+}$ activity at reward retrieval should be higher than in CON mice, but this was not the case. Also, the sub-baseline $Ca^{2+}$

activity after reward retrieval in control mice (see also ref. [49]) was absent in CSS mice. A parsimonious interpretation of these findings is that typical (control) behaviour in the DRLM test requires activation of BA-NAc reward-responsive neurons, and decreased activation of these neurons in CSS mice contributes to deficient DRLM behaviour.

Turning to the REV test, the operant behaviour-DS–US contingency becomes increasingly effortful, and therefore aversive, as the test progresses. CSS mice display deficient REV behaviour, reflecting reduced US salience, increased perceived effort, or both. In CON mice, in the operant phase at PR 5, BA-NAc neural pathway $Ca^{2+}$ activity decreased below baseline, possibly reflecting the attenuated activity of reward-responsive neurons related to effort. In the post-operant phase, CON mouse $Ca^{2+}$ activity increased with PR, from below baseline at PR 5 to above baseline at PR 13; this could reflect increased activity in BA-NAc aversion-responsive neurons related to increased effort. In the post-feeder phase at PR 5, $Ca^{2+}$ activity was highest at reward retrieval/consumption, which could reflect reward neuron activity again. In CSS compared with CON mice, $Ca^{2+}$ activity at PR 5 was high during the long-operant phase and the post-operant phase. A parsimonious interpretation of these findings is that the effort-related activity of aversion-responsive neurons is higher in CSS mice, and this contributes to their deficient REV; quantitatively, $Ca^{2+}$ activity at REV PR 5 in CSS mice corresponds to $Ca^{2+}$ activity at REV PR 13 in CON mice. This finding of increased BA-NAc neural activity associated with the effort in CSS mice is relevant to the human evidence that sensitivity to effort cost is increased (i.e. REV is reduced) in MDD[50]. At reward retrieval, as in the DRLM test, CSS and CON mice displayed similar $Ca^{2+}$ activity peaks, and BA-NAc reward neurons possibly contributed most to this. It has been reported that a substantial proportion of BA pyramidal neurons of undetermined projection respond to both a reward DS and an aversion DS within a single behavioural task;[51] it is possible that some dual-valence BA-NAc neurons contribute to $Ca^{2+}$ activity in both (aversion) post-operant phase and (reward) post-feeder phase of the REV test.

With the aim of identifying further evidence that CSS affects BA-NAc neurons, transcriptome-level expression in population samples of such neurons was investigated. There was no clear separation of CSS and CON mice with respect to expression levels of the 13,000 murine transcripts identified. Indeed, only a moderate number of genes were differentially expressed: almost all of these genes were up-regulated in CSS compared with CON mice; several of these encode proteins involved in either decreasing (*Gata2*) or increasing (*Plxnb2*) spinogenesis at glutamate-neuron dendrites[27,29]. The small number of KEGG pathway gene sets in which up-regulated genes were enriched are related to neuronal signalling, namely Rap1 signalling pathway, MAPK signalling pathway and Axon guidance. At the intermediate BA region studied here, there is an overlap between magnocellular anterior BA (aBA) and parvocellular posterior BA (pBA)[14]. It was reported that *Rspo2* was specific to aBA and aversion neurons, and *Ppp1r1b* to pBA and reward neurons;[15] however, a subsequent study reported that whereas aBA neurons, including those projecting to NAc and responsive to aversion, were *Rspo2*+, pBA neurons expressed both *Rspo2* and *Ppp1r1b*[16]. Both *Rspo2* and *Ppp1r1b* were expressed in the BA-NAc neuron population studied here, and each at similar levels in CSS and CON mice; this probably reflects that the neurons sampled were a mixture of aBA and pBA BA-NAc neurons, although it is not known whether the two genes were expressed by the same or different neurons. The transcriptome also included *Fezf2*, recently described to be a marker gene for BA-NAc aversion neurons and co-expressed with *Rspo2*[16]. In a transcriptome study of amygdala basolateral

complex (BLA) tissue, CSS mice had lower expression of *Ppp1r1b* and various other genes encoding proteins involved in dopamine signalling;[21] that these genes were not dysregulated here indicates that this previous finding involved neurons other than the current study population. Therefore, the main finding that CSS resulted in dysregulated expression of only a moderate number of genes could be due to the heterogeneous neuron population and to CSS exerting specific or even opposing effects on the transcriptomes of BA-NAc reward neurons and aversion neurons.

To investigate whether specific, chronic inhibition of BA-NAc neurons would replicate the CSS effect of reduced DRLM behaviour—coincident with decreased activity in what we hypothesize to be BA-NAc reward neurons—we utilized BA-NAc neuron-specific expression of TeTxLC to induce VAMP2 cleavage and inhibit glutamate release. Indeed, the 15-day expression of TeTxLC led to a marked reduction in the discriminative learning ratio compared with controls, with the learning ratios of TeTxLC mice being remarkably similar to those of CSS mice. These findings are consistent with an attenuation of the DS signal transmitted by BA-NAc reward-responsive neurons to the basal ganglia/mesolimbic reward circuit. That TeTxLC mice obtained a similar number of rewards to control mice indicates that motivation was intact under low-effort conditions. In the REV test, TeTxLC mice displayed a mild reduction in reward-to-effort valuation; that this was moderate compared to that in CSS mice could be due to low reward valuation by inhibited reward neurons combined with and ameliorated by low effort valuation by inhibited aversion neurons. TeTxLC efficacy in terms of VAMP2 cleavage and inhibition of synaptic vesicle fusion and neurotransmitter exocytosis has been reported in terms of low VAMP2 signal at axonal terminals and low amplitude of ex vivo post-synaptic currents[31]. Here we demonstrated decreased VAMP2 immunosignal in the intermediate BA; it was not possible to detect decreased VAMP2 signal in NAc, presumably due to VAMP2 in intact axonal inputs from several NAc afferents and in NAc neuron somata. To measure TeTxLC efficacy in the transfected BA-NAc neuron population per se, transcriptome-level differential gene expression was deployed. The up-regulation of genes in specific KEGG pathway gene sets provides indirect but convincing evidence that chronic VAMP2 cleavage resulted in glutamate accumulation which in turn activated neuronal senescence and apoptotic processes in the BA-NAc neurons as well as increased proximity and activity of glial cells.

To investigate whether specific, chronic activation of BA-NAc neurons would replicate the CSS effect of reduced REV behaviour—coincident with increased activity in what we hypothesize to be BA-NAc aversion neurons—we utilized pathway-specific expression and actuation of the excitatory DREADD rM3D(Gs). Our rationale was to induce, in the BA-NAc neural pathway, low continuous DREADDs activation via CNO in drinking water to simulate the chronic-distal component of CSS, and high acute activation via CLZ injection to simulate the acute-proximal component of CSS. Firstly, in the DRLM test, 15-day DREADDs activation was without effect on the discriminative learning ratio relative to controls, perhaps attributable to a net zero effect of increased activity in BA-NAc reward as well as aversion neurons. DREADDs activation did lead to reduced DS responses/rewards obtained, consistent with a motivational deficit similar to that in CSS mice and possibly reflecting increased effort valuation attributable to BA-NAc aversion neurons. In the REV test, DREADDs mice displayed a moderate reduction in reward-to-effort valuation; this is consistent with high-effort valuation mediated by activated aversion neurons, which is then transmitted to basal ganglia/mesolimbic circuitry. In a previous study of some relevance here[52], when mice were administered corticosterone (CORT) chronically they did not show a decrease in effortful reward responding but did make fewer feeder responses on a low-effort schedule, and DREADDs activation of BA-NAc neurons directly before low-effort behavioural testing increased feeder responses in both chronic-CORT and control mice; the respective findings of the two studies highlight the importance of the manipulation used and the duration of DREADDs activation of the BA-NAc neurons.

The current findings for the inter-relationships between chronic social stress, activity of BA-NAc glutamate neurons, and specific aspects of reward processing and behaviour, and a hypothesis to account for causal mechanisms, are summarized in Fig. 6. The current findings are consistent with there being two major subpopulations of BA-NAc neurons, one of monovalent neurons activated by reward and one of monovalent neurons activated by aversion including effort. Although previous studies have reported that there are more such reward neurons than aversion neurons[12,13,15], a recent study from our laboratory indicates that, for the intermediate BA-NAc at least, the two neuronal subpopulations are of similar size. There is also evidence that BA-NAc reward and aversion neurons can be mutually antagonistic: it has been reported[15] that optogenetic activation of *Rspo2*+ (putative aversion) BA neurons during discriminative reward learning led to decreased reward responding and *Fos* expression in *Ppp1r1b*+ (putative reward) BA neurons[15] (also, optogenetic activation of *Ppp1r1b*+ BA neurons during contextual aversion learning led to decreased conditioned freezing and *Fos* expression in *Rspo2*+ BA neurons). Extrapolating to the current findings, if CSS leads to increased synaptic plasticity in BA-NAc aversion neurons, this could result in: (1) chronic inhibition of BA-NAc reward neuron activity and, as potentially replicated in the TeTxLC experiment, decreased DRLM; (2) increased effort valuation and, as potentially replicated in the DREADDs experiment, decreased REV (Fig. 6). There is evidence that chronic stress leads to hypertrophy of and increased spinogenesis on dendrites of BA neurons including BA-NAc neurons, obtained using chronic restraint stress[19,53,54] or chronic social defeat[55]. Increased inhibition of BA-NAc reward neurons by hyper-trophic and -active BA-NAc aversion neurons would require mediating GABAergic interneurons: BA GABA interneurons function as disinhibitors of aversive conditioning[56,57], rendering their involvement in glutamate neuron inhibition as plausible[15]. Finally, with respect to the differential effects of BA-NAc reward and aversion neurons at NAc medium spiny neurons (MSNs), possibilities include (1) Projection to different glutamate-receptor expressing axonal terminals of ventral tegmental area (VTA) dopamine neurons synapsing in the NAc, with different effects on glutamate release[58,59]. (2) Projection to different NAc MSN subtypes; for example, BA reward neurons might project to DA receptor 1 (D1R) expressing MSNs and BA aversion neurons to D2R MSNs, thereby differentially regulating the contributions of D1R- and D2R-MSNs to basal ganglia/mesolimbic circuitry. (3) Differential projection to NAc MSNs that include heteromeric D1R/NMDA receptor complexes[60]. Of direct relevance here, CSS mice do indeed exhibit attenuated reward-related NAc DA release in both the DRLM and REV tests, as detected using DA sensors and fibre photometry[61].

In summary, this study provides important iterative evidence that chronic social stress leads to task-specific inhibition or activation of BA-NAc neurons, changes associated with and that lead directly to reductions in reward learning and reward-to-effort valuation, respectively. This experimental evidence for the major involvement of the BA-NAc neural pathway in mediating chronic stress effects on specific reward processes also indicates the importance of molecular tools that will enable the separate identification, manipulation and study of BA-NAc reward- and aversion-neuron subpopulations, and of their respective

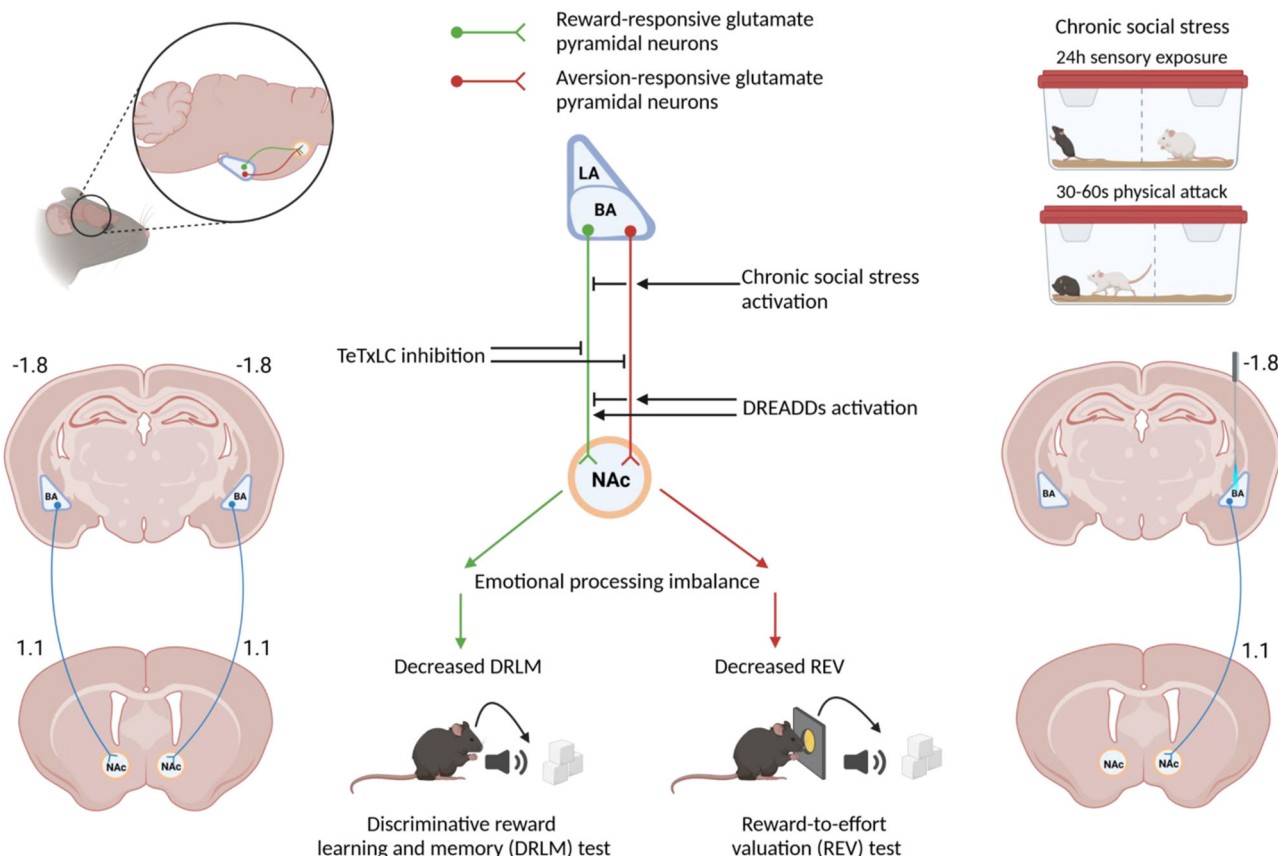

**Fig. 6 Summary of current findings for the inter-relationships between chronic social stress, activity of BA-NAc neural pathway and reward processing, and a proposed hypothesis for causal mechanisms.** Findings: Chronic social stress (CSS) led to reduced reward interest/learning in a discriminative reward learning and memory (DRLM) test and reduced motivation in a reward-to-effort valuation (REV) test, relative to control (CON) mice. The CSS reduction in DRLM co-occurred with decreased BA-NAc neural pathway activity. Chronic inhibition of BA-NAc neural pathway using Cre-dependent TeTxLC led to a CSS-like DRLM reduction and to a mild REV reduction. The CSS reduction in REV co-occurred with increased BA-NAc neural pathway activity. Chronic activation of BA-NAc neural pathway using Cre-dependent excitatory DREADD rM3D(Gs) was without effect on DRLM and led to a CSS-like REV reduction. Hypothesis: Previous studies indicate that the BA-NAc neuron population under investigation comprises intermingled monovalent reward neurons and aversion neurons, with a higher prevalence of reward neurons. A parsimonious interpretation of the current findings is: (1) The BA-NAc subpopulations of reward neurons and aversion neurons are of similar size. (2) CSS decreases the activity of BA-NAc reward neurons and this contributes to reduced DRLM. (3) CSS increases the activity of BA-NAc aversion neurons and this contributes to increased effort valuation and therefore reduced reward-to-effort valuation. (4) CSS-induced increased activity in BA-NAc aversion neurons contributes to decreased activity of BA-NAc reward neurons via BA GABAergic interneurons. Separate identification and studies of BA-NAc reward neurons and aversion neurons will enable testing of the important hypotheses generated by the present study. Images were created with BioRender.com.

projections to D1R- or D2R MSNs in NAc core or shell. Definitive identification of their respective contributions to reward circuitries and processes and chronic stress-induced changes thereof will be important next steps, including in terms of identifying potential molecular targets for novel pharmacotherapies to treat specific reward pathologies.

## Methods

**Animals**. Experiments were conducted with C57BL/6J (BL/6) male mice bred in-house. Mice were weaned with same-sex littermates at age 4 weeks and maintained in littermate pairs from age 5–6 weeks until the end of the experiment unless otherwise stated. Mice were aged 10–12 weeks at the onset of experiments. Mice were maintained in cages measuring (33 × 21 × 14 cm) in an individually ventilated caging system. Temperature was kept at 21–23 °C and humidity at 50–60%, and the light cycle was reversed with lights off at 07:00-19:00 h. Mice were fed with food pellets (Complete pellet, Provimi, Kliba AG, Kaiseraugst, Switzerland) which were provided ad libitum except during behavioural conditioning and testing (see below). Water was provided ad libitum including during behavioural testing. All experimental procedures were conducted during the dark phase and between 09:00 and 17:00 h. Each experiment began with the handling of mice for 5 min per day each on 3-5 consecutive days. All experiments were conducted under an animal experiment license issued by the Veterinary Office of Canton Zurich (ZH-155/2018).

Experiments with the chronic social stress (CSS)-induced attenuated reward processing model that provides the basis for the current study are typically conducted with $n = 12$–14 CSS versus $n = 12$–14 control mice. This sample size yields robust and reproducible CSS effects (e.g. refs. [22,24,25]), and was, therefore, the sample size used for behavioural experiments in the current study. Because we did not have prior data for the effects of CSS on BA-NAc neural activity during reward processing, for the fibre photometry experiment we aimed for a sample size of 18–20 mice per group.

**Chronic social stress (CSS)**. The chronic social stress (CSS) procedure used[20] is based on the resident-intruder paradigm and includes refinements from similar procedures used during chronic social defeat. Resident mice were unfamiliar, aggressive, ex-breeder CD-1 males aged 8–10 months and weighing 38–55 g. Caged singly, on the day prior to the onset of CSS, a transparent, perforated plastic divider was placed along the length of the home cage of each CD-1 mouse, separating the cage into two equal compartments. The CSS mice were derived from pairs of BL/6 littermates that were separated from each other on day 1 of the CSS procedure. A CSS mouse was introduced into the same compartment as a CD-1 mouse for a cumulative total of 60 s physical attack or 10 min maximum. After this acute proximal stressor, the CSS mouse remained in the same compartment and the CD-1 mouse was transferred to the opposite compartment; the CSS mouse was exposed to this visual, olfactory and auditory distal stressor for 24 h. The following day, the CSS—CD-1 mouse pairings were rotated so that each CSS mouse was placed with a different (unfamiliar) CD-1 mouse, firstly for proximal attack and then for distal

exposure, and this continued across days. The total duration of the CSS protocol was 15 days. It is essential that the environmental stressor is not confounded by bite wounds and, in addition to the refinement of timing and restricting daily attacks to 60 s maximum, the lower incisor teeth of CD-1 mice were trimmed every 3 days. The mean cumulative duration of daily attack experienced by CSS mice was 45–50 s; all CSS mice displayed submissive behaviour and vocalization during the proximal stressor. The mice in the control or comparison group (CON) comprised pairs of littermates that were maintained together and handled for 1 min on each of the 15 days.

The effects of CSS were studied in two experiments, namely the effects of CSS on reward-directed behaviour and the effects of CSS on BA-NAc glutamate neuron activity during reward-directed behaviour. From day 15 and throughout behavioural testing, each CSS mouse remained in the same divided cage with the same CD-1 mouse without further attacks.

**Conditioned reward-directed behaviour**. For behavioural experiments without fibre photometry, the apparatus and procedures used are detailed below. For the behaviour-fibre photometry experiment, additional details of apparatus, conditioning and testing are given in the corresponding section. All behavioural conditioning and testing were conducted in infra-red-illuminated rooms adjacent to the mouse holding room.

*Controlled feeding and body weight*. Prior to the onset of conditioning (training), body weight (BW) per mouse and food intake per littermate pair were measured for each 24 h period across 1 week. For each mouse the mean daily value provided its baseline; for food intake per mouse, the respective BWs of the two littermates were taken into account assuming that the heavier mouse would consume proportionately more. Beginning the following week, mice were food restricted so that BW was reduced to 90–95% of baseline; this ensured adequate motivation for conditioning using sucrose pellet reinforcement. On the day prior to the onset of conditioning, mice were familiarized in the home cage with the sucrose pellets to be used as reinforcement. During CSS/CON or AAV vector experiments, BW and food intake were measured to provide re-baseline values (re-BBW, re-B-food intake). During each day of testing of conditioned reward-directed behaviour, mice were carefully food restricted so that BW was 95–100% re-BBW directly prior to test onset. The required amount of normal diet was placed in the home cage 2–3 h after testing, and all food had been consumed prior to the time of testing on each day.

*Apparatus*. Modular conditioning chambers had inner dimensions of $20 \times 17 \times 18$ cm and a house light provided 10 lx illumination; four such chambers, each placed within an attenuation chamber into which background white noise was presented, were run in parallel by a control PC and interface (TSE Systems, Bad Homburg, Germany)[63]. In each chamber, a feeder port ($\varnothing = 20$ mm $\times$ depth $= 35$ mm) was located in the middle of one side wall. Food pellets were delivered singly into the feeder port from a pellet dispenser and could be retrieved by the mouse extending its snout into the feeder (feeder response); each such response into the feeder was detected via an infra-red motion sensor and recorded. A speaker was located above the feeder via which a tone stimulus could be presented. An operant stimulus ($\varnothing = 20$ mm $\times$ depth $= 30$ mm) activated by nose-poke (operant response) could be inserted to the side of the feeder (centre-to-centre distance $= 55$ mm); a white LED set into the recess of the operant stimulus was illuminated to indicate it was active, and operant responses were detected via an infra-red motion sensor and recorded. Water was available from a bottle and spout placed at the wall opposite to the feeder and operant stimulus. The chamber floor and walls were wiped with 70% ethanol between mouse runs.

*Conditioning*. Conditioning sessions were conducted on consecutive days and each had a maximum duration of 30 min. Mice were trained with sucrose pellets (14 mg, F05684 Dustless Precision Pellets, Bio-Serv). All training steps were conducted in the absence of tone stimuli. Firstly, without an operant stimulus in the chamber, mice learned that sucrose pellets were available in the feeder. In the first session, 15 pellets were placed in the feeder at session onset and 1 further pellet was delivered automatically each 45 s; in subsequent sessions, 1 pellet was placed in the feeder at session onset and 1 further pellet was delivered automatically each 45 s, and mice were required to retrieve and eat at least 30 pellets in 1 session; 2–7 sessions were required. At the next stage, mice needed to make a feeder response to trigger pellet delivery (after circa 0.5 s) and the learning criterion was 2 consecutive sessions with at least 30 pellets retrieved and eaten; mice required 2–6 sessions. Then the operant stimulus was introduced, and mice learned that 1 operant response (fixed ratio 1, FR1) into the illuminated port was required to extinguish the LED and trigger pellet delivery; the subsequent feeder response was followed by a 5 s time out and the operant stimulus was then active/illuminated again. In FR1 sessions 1–3, 5, 3 and 1 pellets, respectively, were placed in the operant stimulus, and thereafter no pellet. Mice were required to complete at least 30 FR1 trials and consume at least 30 pellets in 2 consecutive sessions; 3–6 sessions were required. In a final FR1 session, chocolate-flavoured sucrose pellets (20 mg, F05301 Dustless Precision Pellets, Bio-Serv) were used; mice preferred these to the training pellets, and they were the gustatory stimulus used for testing reward-directed behaviour. Mice required 12–15 days to complete the three training stages. For procedures between the

completion of training and onset of behavioural testing, see the sections for the specific experiments.

*Testing*. Behavioural testing was conducted under mild food restriction only (see above) in order to minimize the effect of homoeostatic appetite systems on behaviour and thereby maximize test sensitivity to gustatory reward salience and group differences therein[22,64]. In addition, to control for potential differences in homoeostatic appetite between groups/subjects, in each experiment in either the final DRLM test or the final REV test (see below), a pellet (3 g) of normal food was placed on the chamber floor as a low-effort/low-reward alternative to chocolate pellets. Mice would be expected to consume the normal food in a large amount relative to chocolate pellets only if their feeding behaviour was driven primarily by homoeostatic appetite rather than gustatory reward.

Discriminative reward learning-memory (DRLM) test: The chamber contained the feeder and no operant stimulus. The session was initiated by presenting a novel (neutral) tone at 6.5 kHz and 80 dB; the tone had a maximum duration of 30 s and 1 feeder response triggered chocolate pellet delivery (0.5 s) and tone termination after 1 s. The interval between consecutive tones was $50 \pm 30$ s (inter-trial interval, ITI, on a variable interval schedule). Feeder responses during the ITI were counted but without consequence. Therefore, the tone served as a discriminative stimulus (DS) that signalled when a feeder response is coincident with reward availability; the higher the reward salience, the faster and higher the amount of discriminative learning expected, measured as a relative decrease in response latency during DS compared with ITI. Successive tests allowed for the study of discriminative learning-memory. Per DRLM test, the maximum number of DS–US trials was 40 and session duration was set to 50 min (maximum) to ensure that all mice received 40 trials. Mice were tested on 3 or 4 consecutive days depending on the experiment. Depending on the experiment and group, mice completed 17–33 DS–US trials per test (Table S2). In each test, the first 20 trials were analysed (Test 1: trials 1–20, Test 2: trials 41–60, Test 3: trials 81–100, Test 4: trials 121–140), and the measures of interest were: number of chocolate pellets obtained (=number of trials on which a feeder response was made during the DS); median DS response latency; median ITI response interval (ITI duration (s)/feeder responses per ITI); discriminative learning ratio calculated as median ITI response interval/median DS response latency.

Reward-to-effort valuation (REV) test: The testing of REV began 1–3 days after completion of DRLM testing, depending on the experiment. The chamber now also contained the operant stimulus. The session duration was 45 min and no break point was used. Each test session was initiated with operant stimulus LED illumination, and 1 operant response elicited simultaneous extinguishing of the LED, 1 s tone DS (6.5 kHz, 80 dB) and chocolate pellet delivery into the feeder; feeder response/pellet retrieval was followed by a 5 s time out and then illumination of the operant stimulus LED. A progressive ratio (PR) reinforcement schedule was used as follows: trials 1–5 at PR1, trials 6–10 at PR5, trials 11–15 at PR9, trials 16–20 at PR13, and so on. As for the DRLM test, the REV test measures reward interest, but because reinforcement is on a PR schedule it additionally allows for measurement of reward valuation relative to aversive effort valuation in terms of activity and time required to obtain the reward (www.nimh.nih.gov/research/rdoc). Mice were tested on 2 or 3 consecutive days depending on the experiment. The initial test served as a transition test from the DRLM test conditions, and the data from REV test 2 and/or 3 were used for analysis. The measures of interest were: total number of operant responses; number of chocolate pellets earned; final ratio attained; pellet retrieval latency; post-reinforcement pause.

For further details of the procedures for studying conditioned reward-directed behaviour (see refs. [22,24,25,63,65]).

**Stereotactic surgery**. Stereotactic surgery was conducted according to our previously published protocol (e.g. ref. [66]). Both mice per littermate pair were operated successively on the same day. For analgesia, buprenorphine (Temgesic, 0.1 mg/kg s.c.) was administered 0.5–1.0 h pre-operatively. Mice were anaesthetized using isoflurane in pure oxygen, 4% for induction followed by 1.5–1.75% for maintenance. The mouse was placed in a stereotactic frame (Angle Two™, Leica) and a heating pad was used to maintain body temperature. Ophthalmic ointment was applied to the eyes (Viscotears, Novartis) and disinfectant (Betadine) was applied to the incision site. An incision was made at the cranial midline, and local anaesthetic (lidocaine 10 mg/kg and bupivacaine 3 mg/kg) was applied. Skin and connective tissue were pulled to the sides, and burr holes ($\varnothing = 300$ μm) were drilled into the cranium for injection of viral vectors. Injections of viral vectors were conducted using 10 μl NanoFil™ microsyringes fitted with a 33 G bevelled stainless-steel needle and connected to an ultra-micro pump (UMP3, Micro4, World Precision Instruments), at a rate of 50 nl/min. After completion of an injection the microsyringe remained in position for 5 min and then slowly withdrawn. The burr holes in the cranium were filled with bone wax (Bonewax, Ethicon®) and stainless steel wound clips (Reflex 7, CellPoint Scientific) were applied for wound closure. The mouse was returned to its home cage and remained on a heating pad until it was observed to be active, which required 0.5–1.0 h. Buprenorphine was injected again at 4–5 h post-surgery and administered via the

drinking water for 3 days. Mice were weighed and wound healing was controlled for 10 days post-surgery.

Concerning the stereotactic injection coordinates, in all experiments the coordinates were set to inject into the basal amygdala (BA) at bregma anterior–posterior (AP) −1.8 mm, medial–lateral (ML) ± 3.3 mm, dorsal–ventral (DV) −5.1 mm, and into the nucleus accumbens (NAc) at bregma AP + 1.1 mm, ML ± 1.2 mm, DV −4.8 mm, according to a mouse brain atlas[62]. The coordinates were selected based on in-house evidence and published evidence[12] that the absolute number of BA neurons projecting to the NAc at bregma AP + 1.1 is highest at bregma AP −1.8. The NAc coordinates were selected based on those used in some previous studies[15,17] and to target BA projectors to both NAc core and shell sub-regions.

**Effects of chronic social stress on reward-directed behaviour.** Mice (n = 28) underwent conditioning for behavioural testing and were then returned to unrestricted feeding; they were allocated to CSS (n = 14) and CON (n = 14) groups by counterbalancing on body weight (BW) and number of conditioning sessions, and the CSS/CON protocol began 3 days later. During the CSS/CON procedure (days 1–15), BW and food consumption per CSS mouse and CON mouse-pair were measured on days 1–12; mean values of BW and daily food intake were used as re-baseline values for these parameters. Starting on day 13 and continuing until the last day of behavioural testing, mice were food restricted to be at 95–100% re-baseline BW immediately prior to testing on each day.

The overall mean duration of daily attack received by CSS mice was 50.4 ± 3.8 s (range: 42.1–54.9 s). The baseline and re-baseline data for BW and food intake are given in Table S1. Across CSS/CON days 1–15, the BW of CON and CSS mice remained close to baseline. As expected based on previous studies[22,24,25,42,43], absolute daily food intake was higher in CSS than CON mice during this period. During behavioural testing on days 17–22, CON and CSS mice were food restricted to similar % re-baseline BW values; this required a lower reduction in % re-baseline daily food in CSS compared with CON mice, and absolute daily food intake was higher in CSS mice than CON mice. Mice underwent DRLM tests on days 17–19 and REV tests on days 21–22 with a pellet of normal food available in the test chamber on day 22. The number of DS-US trials with a response per test was lower in CSS mice (Table S2). In the REV test on day 22, there was no difference in the amount of normal food eaten by CSS mice (0.10 ± 0.09 g) and CON mice (0.11 ± 0.09 g) (p = 0.67).

**Effects of CSS on BA-NAc glutamate neuron activity during reward-directed behaviour using fibre photometry**
*Conditioned reward-directed behaviour.* Food restriction and BW regulation were conducted as described above (conditioned reward-directed behaviour). For conditioning and testing, a photometry conditioning chamber running IntelliMaze software (TSE Systems) was used; it had inner dimensions of 20.5 × 26.5 × 26.5 cm (W × L × H) and was fitted with a house light providing 10 lx; it was placed within an attenuation chamber. The chamber contained a feeder located in the centre of one side wall (Ø = 16 mm × depth = 16 mm) and extending 26 mm into the chamber; it was constructed in this way to enable mice fitted with a cranial optic fibre and patch cord to retrieve pellets. Each response into the feeder was detected via an infra-red motion sensor. An enlarged operant stimulus (22 × 26 mm × depth = 32.5 mm) activated by nose-poke (operant response) could be inserted to the left of the feeder on the same side wall; a white LED set into the rear of the operant stimulus was illuminated to indicate it was active, and operant responses were detected via an infra-red motion sensor. The centre-to-centre distance between the operant stimulus and the feeder could be set to 55 mm ("near") or 110 mm ("far"). Water was available from a dispenser placed at the opposite side wall. Reward pellets were delivered from a pellet dispenser directly into the externalised feeder.

Conditioning was conducted as described above with the following specific differences: For sessions 1 and 2, littermates were placed in the chamber together to retrieve and consume sucrose pellets delivered spontaneously into the feeder. In subsequent sessions, mice were placed in the chamber singly and were required to make a feeder response to trigger pellet delivery (circa 0.5 s) and the learning criterion was 2 consecutive sessions with at least 20 sucrose pellets retrieved and eaten. Operant stimulus training comprised sessions with the operant stimulus "near" (2–4 sessions) and then "far" (2–3 sessions), and mice were required to complete at least 20 FR1 trials and consume at least 20 sucrose pellets. Testing was conducted as described above with the following specific differences: The DRLM test was conducted with a tone stimulus of 6.5 kHz and 75 dB presented via a speaker located at the rear wall of the chamber. The maximum number of DS-US trials per test was 35 and the maximum session duration was 35 min. The REV test was conducted with the "far" distance between operant stimulus and feeder and with a maximum session duration of 30 min.

*Adeno-associated viral vectors.* Recombinant AAV vectors (Viral Vector Facility, Zurich Neuroscience Center, ETHZ/UZH) were used to achieve pathway specific expression of the calcium-dependent fluorescent protein GCaMP6 in BA-NAc neurons. A vector encoding *GCaMP6* incorporated into a construct with FLEX, ssAAV-9/2-hSyn1-chI-dlox-GCaMP6m-dlox-WPRE-SV40p(A) ("rAAV-FLEX-GCaMP6"; 4.2 ×10$^{12}$ vg/ml, 300 nl)[1], was injected in the BA. To induce GCaMP6 expression in BA-

NAc neurons specifically, a retrograde vector encoding *Cre-recombinase*, rAAV-2-retro-hsyn1-Cre-mCherry ("rAAV-retro-Cre"; 8.4.10$^{12}$vg/ml)[67], was injected in the ipsilateral NAc (300 nl).

*Stereotactic surgery.* Surgery was conducted as described above with the addition that after AAV vector injection in the BA, a fibre-optic probe (Ø = 400 μm; burr hole Ø = 700 μm) was implanted directly dorsally to the injection site (bregma AP −1.8 mm, ML ± 3.2 mm, DV -4.9 mm). Stable adhesion of the fibre-optic probe onto the cranium was achieved as described previously[66].

*Modified chronic social stress.* During the daily attack sessions, the central divider was removed from the cage to avoid the optic fibre becoming caught in the divider perforations. Otherwise, the standard protocol (see above) was used.

*Fibre photometry and BA-NAc glutamate neuron activity.* Fibre photometry for optical recording of neural activity in freely moving mice was conducted as described previously[66]. Briefly, a laser as excitation light source, a high-sensitivity photodetector, and customized software for signal processing, were used. To provide excitation, a 488 nm laser light was focused into a fibre patch cord and delivered at the optic fibre tip in the BA. Openings in the centre of the ceilings of the attenuation and operant chambers allowed for unrestricted movement of the patch cord. The latter was connected to the optic fibre ferrule on the mouse cranium via a ceramic sheath. Back-propagated GCaMP6 fluorescence was focused on a photomultiplier detector and custom-written software code was used for data acquisition (LABVIEW, 2020). Fibre photometry data were analysed using MATLAB. Feeder response, operant response and tone onset generated TTL signals that were recorded simultaneously with the photometry signal. Optical signal data were demodulated at 970 Hz and down sampled to a sampling frequency of 20 Hz.

*Experimental design.* Mice (n = 48) underwent behavioural conditioning under food restriction. They were then returned to unrestricted feeding and underwent stereotactic surgery. At days 13-14 post-surgery, in order that they experienced operant responding and sucrose pellet retrieval with the patch cord attached to the optic fibre prior to testing, mice were given a conditioning session with operant stimulus present (i.e. REV test condition) and the following day with operant stimulus absent (i.e. DRLM test condition). Mice were then allocated to CON and CSS groups by counterbalancing on BW and required conditioning sessions, and underwent the modified CSS/CON protocol (days 1–15). During days 1–12 of the CSS/CON protocol, BW and food consumption per CSS mouse and CON mouse-pair were measured daily; the mean values of BW and daily food intake were used as re-baseline values for these parameters. Starting on day 13 and continuing until the last day of behavioural testing, mice were food restricted to yield 95–100% re-baseline BW immediately prior to each behavioural test session. On day 16 mice were placed in the conditioning chamber without any stimuli and connected to the patch cord: the GCaMP6 photometry signal of each mouse was recorded for 15 min to check for a sufficient and stable signal, and mice with no/low signal relative to background were removed from the experiment. Mice underwent a DRLM-fibre photometry test on each of days 17–19 and a REV-fibre photometry test on each of days 20–22.

The overall mean duration of daily attack received by CSS mice was 52.8 ± 7.3 s (range: 32.5–60.0 s). The baseline and re-baseline data for BW and food intake are given in Table S1. Across CSS/CON days 1–15, the BW of CON and CSS mice remained close to baseline. As expected, based on previous studies[22,24,25,42,43], absolute daily food intake was higher in CSS than CON mice during this period. During behavioural testing on days 17–22, CON and CSS mice were food restricted to similar % re-baseline BW values and absolute daily food intake was higher in CSS mice than CON mice.

For the DRLM-fibre photometry test, the number of DS-US trials with a response per test was lower in CSS mice than CON mice (Table S2). The first 20 DS–US trials of each test were analysed; as described above, behavioural measures of interest were number of chocolate pellets obtained, median DS response latency, median ITI response interval and learning ratio. For analysis of BA-NAc neuron population activity, for each of trials 1–20, they were categorized as trials with response or without response. Each such trial with response was analysed individually and was subdivided into the following phases: The 10 s prior to DS onset was the trial baseline phase. From DS onset until a feeder response was the DS-on phase and this phase was time normalized and divided into 10 equivalent intervals. Time normalization was conducted according to Yoshida et al.[68]: it refers to the method of fixing a time phase that is of interest and variable length to one standard size of arbitrary units; in addition, the time-normalized period can be divided into n equal intervals of arbitrary duration. From feeder-response onset until 5 s had elapsed was the post DS-feeder phase and was divided into 10 × 0.5 s intervals. After the end of a post-DS feeder phase, the first feeder response indicated the onset of the ITI feeder phase which lasted for 5 s and was divided into 10 × 0.5 s intervals. For each trial with a response in trials 1–20, during the DS-on phase, post-DS feeder phase, or ITI feeder phase, for each 0.05 s time bin (t), the z-scored (normalized) signal intensity (F) was calculated using the formula $(F(t)-F_0)/SD_0$, where $F_0$ and $SD_0$ denote mean and standard deviation of baseline phase activity. The mean z-scored $F(t)$ for trials with response in trials 1–20 was

calculated for each t and each test and mouse. These mean z-scored signal $F(t)$ values were then binned into time-normalized intervals or 0.5 s intervals for statistical analysis[66].

For the REV-fibre photometry test, as described above, the major behavioural measures of interest were total number of operant responses, number of chocolate pellets earned, and final ratio attained. For analysis of BA-NAc neuron population activity, trials were grouped and analysed according to the progressive ratio (e.g. PR 5, PR 9) to which they pertained. Each trial was divided into the following phases: The 10 s prior to the first operant response was the trial baseline phase. From operant response 1 until the final operant response required to reach the current PR was the operant phase; it was time normalized and divided into 10 equivalent intervals. From the final operant response and the 1-s DS that it elicited until the feeder response was the post-operant phase; it was time normalized and divided into 10 equivalent intervals. From feeder-response onset until 5 s had elapsed was the post-feeder phase and was divided into $10 \times 0.5$ s intervals. For each completed trial at PR 5, PR 9 or PR 13, during the operant phase, post-operant phase or post-feeder phase, for each 0.05 s time bin ($t$), the normalized (z-scored) signal intensity ($F$) was calculated using the formula $(F(t)-F_0)/SD_0$. The mean z-scored $F(t)$ for completed trials at PR 5, PR 9 or PR 13 was calculated for each $t$ and each test and mouse. These mean z-scored signal $F(t)$ values were then binned into time-normalized intervals or 0.5 s intervals for statistical analysis.

*Fibre photometry signal testing and target validation.* Five mice were excluded from behavioural-fibre photometry testing due to no or low GCaMP6 signal at signal testing on day 16 (CON $n = 3$, CSS $n = 2$). In the REV test specifically, 4 mice were excluded due to technical failures (CON $n = 1$, CSS $n = 3$). After completion of behaviour-fibre photometry testing, mice were deeply anaesthetised and underwent brain perfusion-fixation for histological assessment of the accuracy of GCaMP6 BA-NAc neuron targeting in terms of BA probe placement and BA GCaMP6 and NAc mCherry expression. As described in detail elsewhere[66], the optic fibre implant was removed and the brain was sectioned coronally at 100 μm using a vibratome (Leica). Sections underwent Nissl staining (NeuroTrace 640/660 Deep-Red Fluorescent Nissl Stain, Thermo Fisher), followed by washing in PBS, mounting on microscope slides, addition of Dako/DAPI fluorescence mounting medium (Sigma Aldrich), and cover-slipping. Using an epifluorescence microscope (Axio Observer.Z.1, Zeiss), mounting medium allowed for localization of GCaMP6 expression, and Nissl staining allowed for localization of the optic fibre placement and tracing of the BA–NAc pathway using mCherry reporter. Using a mouse brain atlas[62] the bregma level of the BA section that included the most ventral position of the fibre tip in the BA, and the NAc section with the highest mCherry expression, were identified. One mouse (CON) was excluded due to a misplaced optic fibre. Supplementary Fig. 1a provides representative examples of histological verification of GCaMP6 and optic fibre tip placement in BA, and NAc mCherry expression. Supplementary Fig. 1b provides the estimated BA location of the optic fibre tip and GCaMP6 expression in CON and CSS mice, based on histological assessments. Supplementary Fig. 1c provides the estimated area of mCherry expression in the NAc.

**Effects of CSS on the transcriptome of BA-NAc glutamate neurons**

*Experimental design.* Mice ($n = 32$) were allocated at random to CON ($n = 12$) and CSS ($n = 20$) groups; more mice were allocated to the CSS group to allow for the possibility that variance in the BA-NAc neuron population transcriptome would be higher in this group. Mice underwent stereotactic surgery to inject the retrograde neuronal tracer cholera toxin subunit β conjugated with Alexa Fluor™ 555 (CTB-555, Invitrogen) into the NAc in each hemisphere in a volume of 300 nl. Beginning at 7 days post-surgery, mice underwent the CSS/CON protocol (days 1–15). The overall mean duration of daily attack received by CSS mice was $46.4 \pm 1.2$ s (range: 43.9–48.6 s). Body weight (mean CSS/CON days 1–15) was lower in CSS mice ($28.5 \pm 1.2$ g) than CON mice ($31.3 \pm 1.2$ g: $p < 0.001$); the efficacy of the CSS protocol was demonstrated by increased weight of adrenal glands (CSS $5.2 \pm 0.7$ mg, CON $3.4 \pm 0.7$ mg; $p < 0.001$) and spleen (CSS $94.1 \pm 23.6$ mg, CON $71.6 \pm 4.0$ mg; $p < 0.03$) in CSS compared with CON mice, as measured at day 16. On day 16, mice were deeply anaesthetized and then perfused with PBS at RT for 1 min. The brain was removed and placed in a cryo-mould (E6032-1CS, Sigma) with embedding medium (Tissue-TEK OCT Compound). The cryo-mould was then placed on powdered dry ice for 10 min, wrapped in aluminium foil, placed in a polythene bag and stored at −80 °C, prior to laser capture microdissection (see BA-NAc glutamate neuron population transcriptomics). In CON and CSS mice, 3 and 4 subjects, respectively were excluded due to insufficient CTB+ BA neurons at AP −1.6 to −2.0 mm, yielding final sample sizes of CON = 9 and CSS = 16.

**Effects of tetanus toxin light chain inhibition of BA-NAc glutamate neurons on reward-directed behaviour**

*Adeno-associated viral vectors.* The rAAV vectors were constructed and produced by the Viral Vector Facility, Zurich Neuroscience Center, ETHZ/UZH. A Cre-Lox inversion recombination rAAV system was used to achieve expression of the tetanus toxin light chain protein (TeTxLC) specifically in BA neurons projecting to NAc. TeTxLC cleaves the synaptobrevin vesicle-associated membrane protein 2 (VAMP2) involved in docking-fusion of synaptic vesicles in glutamate neurons and thereby reduces synaptic glutamate release[31,69]. An rAAV vector encoding *TeTxLC* incorporated into a construct with a double-floxed inverted open reading frame

(flip-excision genetic switch, FLEX), rAAV-9/2-hSyn1-dlox-TeTxLC(rev)-dlox-eGFP ("rAAV-dlox-TeTxLC-eGFP"; titre $2 \times 10^{11}$ vg/ml), was injected in the BA (300 nl per hemisphere). To induce rAAV-TeTxLC expression in BA-NAc neurons specifically, the retrograde vector rAAV-retro-Cre (300 nl; see fibre photometry experiment) was injected in the NAc. rAAV-2-retro-hsyn1-mCherry ("rAAV-retro-mCherry"; $5.6 \times 10^{12}$ vg/ml; 300 nl) was injected in the NAc as a control vector. To investigate whether rAAV-TeTxLC at its original titre ($5.7 \times 10^{12}$ vg/ml) impacted on amygdala function even in the absence of expression induction, a pilot study was conducted with Pavlovian aversion learning-memory, an amygdala-dependent paradigm. Mice underwent an activity test (no tone or footshock) in the conditioning context on day 1, a tone-footshock (CS–US) conditioning test on day 2 (20 s tone with seconds 19–20 contiguous with 0.2 mA footshock), and a tone expression test on day 3. For details of the apparatus and test protocol (see ref. [24]). Mean percent time spent freezing was quantified during CS presentations and/or intervals in each test phase. Aversive memory expression was inhibited by the original titre whilst there was no effect, neither in the absence nor the presence of Cre-Lox recombination, at the 1:25 diluted titre of $2 \times 10^{11}$ vg/ml, which was then used in the main experiment (supplementary Fig. S4a–d).

*Experimental design.* Mice ($n = 36$) underwent behavioural conditioning and were allocated to experimental groups based on BW and required conditioning sessions. Mice then underwent stereotactic surgery, with both littermates allocated to the same group: Mice in the experimental group, BA TeTxLC × NAc retro-Cre ($n = 12$), received rAAV-TeTxLC bilaterally in BA and rAAV-retro-Cre bilaterally in NAc. Mice in the control groups were injected bilaterally with either rAAV-TeTxLC in BA and rAAV-retro-mCherry in NAc (BA TeTxLC × NAc retro-mCherry), or phosphate-buffered saline (PBS) vehicle (VEH) in BA and rAAV-retro-Cre in NAc (BA VEH × NAc retro-Cre); to enable identification of the injection site, both control groups were co-injected in the BA with cholera toxin subunit β conjugated to Alexa Fluor 488 (CTB-488; Invitrogen), with rAAV vector titre and volumes adjusted to maintain parity with the experimental group. Beginning 5 days post-surgery, BW and food intake were measured daily for 8 days, and the mean values were used as re-baseline values for these parameters. The baseline and re-baseline data for BW and food intake are given in Table S1; there were no group differences, and this was also the case during behavioural testing. Starting at 14 days post-surgery (designated as day 14) and continuing until the last day of behavioural testing, mice were food restricted to yield 95–100% re-baseline BW immediately prior to each test session. Behavioural testing commenced on day 15, and comprised 4 daily DRLM tests on days 15–18 (Table S2) with a pellet of normal food provided on day 18, a 1 day interval, and then 2 daily REV tests on days 20–21. In the DRLM test on day 18, there was no difference in the amount of normal food eaten by mice in the 3 groups ($0.05 \pm 0.07$; Supplementary Fig. 4h).

After completion of testing, mice were euthanized with brain perfusion-fixation for histological verification of rAAV injection sites (see the section "Brain histology, injection site validation and immunofluorescence staining") and VAMP2 immunofluorescence and integrated density analysis. One mouse in the control group BA VEH × NAc retro-Cre was excluded due to misplaced BA injections.

*VAMP2 immunofluorescence and integrated density analysis.* VAMP2 integrated density in BA–NAc neurons was analysed as a marker of TeTxLC cleavage efficacy. In the 12 TeTxLC mice and 6 mice from each control group, two coronal sections at bregma −1.8 mm underwent immunofluorescence staining using rabbit anti-VAMP2 as primary antibody (1:1000; Ab3347, Abcam) and Cy5 goat anti-rabbit (1:500; A10523, Invitrogen) as secondary antibody. For protocol details, see "Brain histology, injection site validation and immunofluorescence staining". For each hemisphere per section, an image comprising the entire BA was acquired using a confocal laser scanning microscope at ×20 magnification (for details, see the section "Microscopy"). Separate laser channels were used for VAMP2 (647 nm), eGFP (488 nm), mCherry (555 nm) and DAPI (405 nm). In control mice and TeTxLc mice, respectively, mCherry and EGFP were used to identify BA-NAc neurons and the area where the fluorescent signal was most intense was used for VAMP2 quantification. Working with the images and Fiji ImageJ, representative images were used to create a region of interest (ROI) that corresponded to the typical area covered by fluorescently labelled BA-NAc neurons. This same ROI was then applied to each BA and its mean integrated density for VAMP2 was calculated as the product of mean grey value × area.

*RNA-Seq validation of tetanus toxin light chain inhibition of BA-NAc neurons.* To further validate the efficacy of TeTxLC cleavage of VAMP2, the effect on the transcriptome expression of BA-NAc neurons was investigated. Naïve mice ($n = 12$) underwent stereotactic surgery and were allocated to either the experimental group BA TeTxLC × NAc retro-Cre ($n = 6$, BW = $31.3 \pm 1.5$ g) or the control group BA TeTxLC × NAc retro-mCherry ($n = 6$, BW = $33.0 \pm 3.2$ g). At 15 days post-surgery mice were deeply anaesthetised and perfused with PBS at RT for 1 min. The brain was removed and placed in a cryo-mould (E6032-1CS, Sigma) containing embedding medium (Tissue-TEK OCT Compound, Sysmex). The cryo-mould was then placed on powdered dry ice for 10 min, wrapped in aluminium foil, placed in a polythene bag and stored at −80 °C, prior to laser capture microdissection (see the section, "BA-NAc glutamate neuron population transcriptomics").

## Effects of DREADDs activation of BA-NAc glutamate neurons on reward-directed behaviour

*Adeno-associated viral vectors.* Recombinant AAV vectors (Viral Vector Facility, Zurich Neuroscience Center, ETHZ/UZH) were used to achieve pathway specific expression of designer receptors exclusively activated by designer drugs (DREADDs). A vector encoding the excitatory DREADD *rM3D(Gs)* incorporated into a construct with FLEX, ssAAV-9/2-hSyn1-dlox-rM3D(Gs)-EGFP(rev)-dlox-WPRE-hGHp ("ssAAV-rM3D(Gs)"; $6.4 \times 10^{11}$ vg/ml, 300 nl) was injected in the BA in both hemispheres. To induce ssAAV-*rM3D(Gs)* expression in BA-NAc neurons specifically, the retrograde vector rAAV-retro-Cre ($8.4 \times 10^{12}$ vg/ml, 300 nl; see tetanus toxin light chain inhibition experiment) was injected in the NAc in both hemispheres. rAAV-retro-mCherry ($5.6 \times 10^{12}$ vg/ml, 300 nl) was used as a control rAAV in the NAc. The original titre of ssAAV-rM3D(Gs) ($6.4 \times 10^{12}$ vg/ml) was demonstrated to affect amygdala-dependent learning (freezing behaviour in Pavlovian aversion learning-memory) in the absence of Cre-recombination, whilst the diluted titre of $6.4 \times 10^{11}$ vg/ml was without non-specific effects (Supplementary Fig. 8a–e).

*Pilot studies to establish actuator doses and DREADDs efficacy.* In the main experiment, both clozapine (CLZ) and CLZ-N-oxide (CNO) were used for DREADDs actuation, and pilot experiments were conducted to establish appropriate study doses. Clozapine (C6305, Sigma-Aldrich) was prepared freshly each day in 0.9% saline with 0.1 N HCl and adjustment to pH 5.5 with sodium carbonate (vehicle). CNO (BML-NS105-0025, Enzo-Life Science) was dissolved in 0.9% saline (vehicle) to give a stock solution of 5 mg/ml, which was aliquoted and stored at −20 °C; 1 aliquot was thawed and diluted in drinking water each day. For CLZ, mice ($n = 6$) not expressing DREADDs were each injected with vehicle, 0.1, 0.3 and 1 mg/5 ml/kg i.p. using a latin-square design and a 2 day washout period between CLZ doses. Immediately after injection, mice were placed in an automated activity test (Multiconditioning System, TSE Systems) for 60 min divided into $12 \times 5$ min blocks and total distance moved was measured. Some mice became more active and other mice less active at 1 mg/kg, whereas at 0.3 mg/kg the activity of all mice was similar to activity under vehicle (supplementary Fig. 8f–h); 0.3 mg/kg CLZ was used in the main study. For CNO, mice ($n = 12$) that had been injected 15 days previously with BA rM3D(Gs)×NAc retro-Cre were injected with 0.1, 0.5 or 2.0 mg/5 ml/kg i.p., and immediately after injection were placed in an open field ($50 \times 50 \times 40$ cm) for 60 min divided into $12 \times 5$ min blocks, and total distance moved was measured. There was no effect of any CNO dose on distance moved (Supplementary Fig. 8i, j); 0.3 mg/kg (administered via the drinking water) was used in the main study.

Next, a pilot study was conducted to confirm the efficacy of BA-NAc neuron rM3D(Gs) actuation by CLZ. Mice expressing the experimental vectors BA rM3D(Gs)×NAc retro-Cre ($n = 3$) or the control vectors BA rM3D(Gs)×NAc retro-mCherry ($n = 3$) were injected at day 15 post-surgery with CLZ at 0.3 mg/kg; after 105 min, mice were deeply anaesthetised and brains perfused-fixed with 4% paraformaldehyde for subsequent immunofluorescence staining and quantification of c-Fos in EGFP$^+$ or mCherry$^+$ BA-NAc neurons (see "Brain histology, injection site validation and immunofluorescence staining").

*Experimental design.* In the main study, mice ($n = 36$) underwent behavioural conditioning and were allocated to experimental groups by counterbalancing on body weight (BW) and number of conditioning sessions. Mice then underwent stereotactic surgery, with both littermates allocated to the same group: Mice in the experimental group received BA rM3D(Gs)×NAc retro-Cre ($n = 12$, $27.6 \pm 2.2$ g) and CLZ/CNO; mice in control groups received either BA rM3D(Gs)×NAc retro-mCherry ($n = 12$, $27 \pm 2.0$ g) and CLZ/CNO, or BA rM3D(Gs)×NAc retro-Cre ($n = 12$, $27.7 \pm 1.5$ g) and vehicle/drinking water. To establish daily water consumption, and therefore the required concentration of CNO in drinking water to achieve an average intake of 0.3 mg/kg/day, on days 6-14 post-surgery daily water intake was measured per littermate pair and the overall daily mean was calculated. Mice drank approximately 8 ml water/cage/day and a CNO solution of 1.5 µg/ml sterile water was provided to all mice that would receive CNO. Beginning at 15 days post-surgery, designated as day 1, mice underwent the following: Experimental mice—BA rM3D(Gs)×NAc retro-Cre CLZ/CNO—received CLZ at 0.3 mg/kg i.p. and CNO in the drinking water at 0.2 mg/kg/day on days 1–24 (adding CNO to the water resulted in mice drinking about 30% less water so that the daily dose of CNO was about 30% less than planned). Control mice BA rM3D(Gs)×NAc retro-mCherry received CLZ and CNO as described for experimental mice. Control mice BA rM3D(Gs)×NAc retro-Cre received vehicle and drinking water on days 1–24. On behavioural test days, mice received CLZ or vehicle at 15 min before test onset. On days 16–19, mice underwent the DRLM test (plus normal food pellet on day 19) and on days 22–24 mice underwent the REV test.

Mean baseline BW of mice was similar in the three groups, as was post-surgery re-baseline BW and mean daily food intake (Table S1). During behavioural testing, mice in control group BA rM3D(Gs)×NAc retro-mCherry CLZ/CNO had a higher % re-baseline BW value than mice in control group BA rM3D(Gs)×NAc retro-Cre VEH, with the experimental group similar to both control groups (Table S1). Intake of CNO in drinking water on days 1–24 was similar in BA rM3D(Gs)×NAc retro-Cre CLZ/CNO mice and BA rM3D(Gs)×NAc retro-mCherry CLZ/CNO mice; BA rM3D(Gs)×NAc retro-mCherry CLZ/CNO control mice drank less than BA rM3D(Gs) x NAc retro-Cre VEH/water control mice (supplementary Fig. 9a).

The % re-baseline daily food intake during behavioural testing was lower in control group BA rM3D(Gs)×NAc retro-mCherry CLZ/CNO than in the experimental group and the other control group, as was the absolute amount of normal food given per day (Table S1). In the DRLM test on day 19, there was no difference in the amount of normal food eaten by mice in the three groups (Supplementary Fig. 9b).

On the day after completion of behavioural testing, some mice were deeply anaesthetised and underwent brain perfusion-fixation for histological verification of rAAV injection sites. In 6 experimental mice and 6 BA rM3D(Gs)×NAc retro-mCherry CLZ/CNO control mice, CNO administration was continued, and at 7 days after completion of behavioural testing these mice received CLZ at 0.3 mg/kg and after 105 min brain perfusion-fixation was conducted. c-Fos immunostaining in BA-NAc neurons was used as a marker to determine whether DREADDs activation was still occurring in the experimental group after chronic CLZ/CNO administration. Per mouse brain, three coronal sections at bregma −1.8 mm underwent c-Fos immunofluorescence staining using a primary antibody of rabbit anti-c-Fos (1:500; 9F6, #2250, Cell Signalling) and a secondary antibody of cyanin-5 goat anti-rabbit immunoglobulin (1:500; A10523 Invitrogen). For protocol details, see Brain Histology. For each hemisphere per section separately, an image comprising the entire BA was acquired using a confocal laser scanning microscope at ×20 magnification (for details, see the section "Microscopy"). Separate laser channels were used for c-Fos (647 nm), eGFP (488 nm) and mCherry (555 nm). In control group mice and experimental group mice, mCherry and eGFP, respectively, was used to identify BA-NAc neurons. From the images, a standard ROI covering most of and specific to the BA region was demarcated and using a custom-written Fiji macro, the numbers of EGFP$^+$, mCherry$^+$, c-Fos$^+$, EGFP$^+$/c-Fos$^+$ and mCherry$^+$/c-Fos$^+$ neurons were obtained, and the mean numbers per hemisphere and per subject were calculated.

## Brain histology, injection site validation and immunofluorescence staining

*Histology.* Mice were deeply anaesthetized by injection of pentobarbital (150 mg/kg i.p.) followed by transcardial perfusion-fixation with phosphate-buffered saline (PBS, pH 7.4, 20 ml) and then freshly-prepared ice-cold paraformaldehyde (PFA, 4% in 0.4 M sodium phosphate buffer, 60 ml). Brains were extracted and placed in 4% PFA for post-fixation for 2 h (VAMP2 in BA-NAc neuron tetanus toxin light chain inhibition experiment) or 5 h (c-Fos in BA-NAc neuron DREADDs activation experiment), and then transferred into 30% sucrose solution for 48 h before being frozen on powdered dry ice and stored at −80 °C. Using a microtome (Leica) at −30 °C, brains were sectioned coronally at 40 µm at bregma 1.6-0.6 mm for NAc sections and −1.2 to −2.4 mm for BA sections. A mouse brain atlas[62] was used to identify sections at the appropriate bregma levels. The sections were stored in tissue collection solution (TCS; glycerine and ethylene glycol in 0.2 M phosphate buffer) (Sigma-Aldrich) and at −20 °C. Using a 24-well plate, sections were placed free-floating in Tris–Triton buffer (pH 7.4) and then underwent immunofluorescence staining (see sections above). After staining, sections were washed in PBS, mounted on microscope slides (SuperFrost, Epredia), Dako/DAPI fluorescence mounting medium (Sigma Aldrich) was added, and the sections cover-slipped for microscopy.

*Microscopy.* For quantification purposes, images were acquired using an inverted confocal laser scanning microscope (Leica SP8) fitted with an HC PL APO CS2 ×20 NA 0.75 multi-immersion objective in standard mode. The pinhole was set to 1 AU (Airy units), pixel size $0.500 \times 0.500$ µm$^2$, and z-stack step size = 2 µm. Following the acquisition of the z-stack images, quantification of VAMP2 integrated density or c-Fos$^+$ cell number was conducted.

For validation of target injection sites in the NAc and/or BA, images of complete coronal sections were acquired using a light epifluorescence microscope (Axio Observer. Z.1, Zeiss), ×5 magnification, 0.55 NA, and 26 mm WD, in order to visualize fluorescent reporter protein expression.

## BA-NAc glutamate neuron population transcriptomics

*Laser capture microdissection.* In the CSS—BA-NAc neuron transcriptome experiment and the BA-NAc neuron tetanus toxin inhibition validation experiment, the readout was differential gene expression in BA-NAc neuronal populations.

Frozen brains were processed using RNA- and RNAse-free conditions throughout. Using a cryostat set at −17 to −20 °C, coronal sections that included the NAc at AP 1.6-0.5 mm were cut at 40 µm and mounted on to microscope slides (SuperFrost, Epredia) and processed to estimate the rAAV injection site (see the section "Brain histology, injection site validation and immunofluorescence staining"). Coronal sections that included the BA at AP −1.6 to −2.0 mm were cut at 10 µm and mounted (three sections per slide) on RNAse-free PET membrane slides (50102, Molecular Machines & Industries, MMI). Sections then underwent fixation and dehydration using the following steps: 70% ETOH at −20 °C for 120 s, dipping in/out ddH$_2$0 ice-cold for 60 s, dipping in/out 70% ETOH ice-cold for 60 s, 95% ETOH ice-cold for 45 s, 100% ETOH at RT for 120 s, 3× washing in xylene at RT, xylene at RT for 120 s. Slides/sections were placed on their edge in a covered dark box at RT for 10 min or until completely dried, and then in a capped 50 ml Falcon tube for either immediate cell collection or stored at −80 °C for 7 days maximum.

Fluorescent reporter protein-labelled tissue was collected from coronal sections at bregma AP −1.6 to −2.0 using a laser capture microdissection system (CellCut, MMI). Fluorescence settings were optimized for visualization of CTB-555$^+$ tissue (channel TRITC) in the CSS—BA-NAc neuron transcriptome experiment, and for visualization of EGFP$^+$ tissue at wavelength 510 nm (channel FITC) or mCherry$^+$ tissue at wavelength 570 nm (channel TRITC) in the BA-NAc neuron tetanus toxin experiment. The membrane slide was positioned and using ×40 magnification, BA tissue areas that were CTB-555$^+$/EGFP$^+$/mCherry$^+$ were each encircled at Ø = 35 μm using the MMI CellTools software. There were 40–60 such areas per BA hemisphere and these were encircled for both hemispheres for each of the three sections on the membrane slide. A MMI Universal UV laser (355 nm, 2 μJ, 4 kHz frequency with 500 pico-s pulse-duration) at 40–50% laser power was activated (velocity = 54 μm/s, focus = 27 μm) and the designated tissue areas were collected on the adhesive cap of an MMI Isolation tube (0.5 ml). The procedure was conducted with 2 membrane slides (6 sections) and isolation tubes per mouse, giving a total of 500–510 tissue samples per mouse. Following tissue collection, tissue lysis was conducted by adding RLT buffer with DTT (100 μl; CSS—BA-NAc neuron transcriptome experiment) or QIAzol® (100 μl/tube; BA-NAc neuron tetanus toxin experiment) to the tube, triturating the tissue on the cap with 20 μl volumes and returning this volume to the tube. The tube was closed, inverted for 15 min at RT, and vortexed for 1 min, inverted for 5 min and centrifuged for 5 s. The tube was then sealed with Parafilm and frozen at −80 °C until RNA extraction.

*RNA isolation and quality control.* Lysate aliquots (2–3 × 100 μL per sample) from the same sample were pooled, and if required RLT or QIAzol was added, to give a final lysis volume of 300 μL. Samples were transferred to 2 ml PhaseLock tubes (QuantaBio). For RLT lysates specifically, a half volume of phenol:chloroform:isoamyl alcohol (25:24:1 v:v:v; Sigma) was added before brief shaking and centrifuging at 4 °C. For both RLT and QIAzol samples, a half volume of chloroform:isoamyl alcohol (24:1 v:v) was added before shaking, a 3 min RT incubation and centrifugation at 4 °C. The aqueous phase was then transferred to a 1.5 ml Eppendorf tube and mixed with a 1.5 volume of isopropanol (Sigma). After thorough pipette mixing, the isopropanol mixture was applied to an RNeasy MinElute spin column and total RNA was extracted using the miRNeasy Micro Kit (Qiagen) with a DNase treatment. Samples were eluted in 14 μl nuclease-free water. RNA samples were assessed both quantitatively and qualitatively using the High Sensitivity Total RNA 15nt Analysis DNF-472 Kit on a 48-channel Fragment Analyser (Agilent). Total RNA yield was 0.31 ± 0.11 ng (mean ± SD) in the RLT lysate samples and 1.45 ± 0.51 ng in the QIAzol lysate samples; RNA integrity could often not be computed due to low input.

*Low input RNA sequencing with poly(A) enrichment.* Up to 1.6 ng of total RNA was used for cDNA synthesis, conducted with the SMART-Seq® v4 Ultra Low Input RNA kit (Takara Bio). Depending on the input amount, 12-15 amplification cycles were conducted. After clean-up, up to 10 ng of cDNA was used to generate the final sequencing libraries with the tagmentation-based DNA Prep Kit (#20018705) and the IDT® DNA/RNA UD Indexes Set A (#20026121), both Illumina®. The index PCR was conducted with 9 or 10 cycles depending on cDNA amount, while the final library was eluted in 30 μL EB Buffer. Low input mRNA libraries were then quantified using the High Sensitivity dsDNA Quanti-iT Assay Kit (ThermoFisher) on a Synergy HTX (BioTek). Library molarity averaged at 21 nM. Libraries were also assessed for size distribution and adapter dimer presence (<0.5%) by the High Sensitivity NGS Fragment DNF-474 Kit on a 48-channel Fragment Analyser (Agilent). All sequencing libraries were then normalized on the MicroLab STAR (Hamilton), pooled and spiked in with PhiX Control v3 (Illumina). The library pools was subsequently clustered on an SP Flow Cell and sequenced on a NovaSeq 6000 Sequencing System (Illumina) with dual index, paired-end reads at 2 ×100 bp length (Read parameters: Rd1: 101, Rd2: 10, Rd3: 10, Rd4: 101), reaching an average depth of 26 million Pass-Filter reads per sample (18.2% CV).

*Differential gene expression and pathway analysis.* Sequencing reads were mapped to the *Mus musculus* reference genome (mm10) using STAR v2.5.2b allowing for soft clipping of adapter sequences. An average of 20 million reads per sample was obtained, from which approximately 10 million reads were assigned to genomic features. Transcript quantification was conducted with RSEM v1.3.0 and feature-Counts v1.5.1. QC, and downstream bioinformatic analyses were conducted with R v4.1.0 and Bioconductor v3.12 tools. Briefly, we identified expressed genes based on the distribution of median log2 raw counts across samples, and this yielded a median of 13,637 expressed genes per sample in the CSS—BA-NAc neuron transcriptome experiment and a median of 13,419 expressed genes per sample in the BA-NAc neuron tetanus toxin experiment. A Gaussian mixture model was fitted to the distribution with mclust v5.4.7 to identify two clusters: genes with median expression values belonging to the cluster with the mean closest to 0 were filtered out from the expression matrix. Then, we normalized the expression matrix using the variance stabilizing transformation from package DESeq2 v1.32.0 and identified the 500 highest variable genes (HVGs). Principal component analysis (PCA) was conducted with these 500 HGVs using PCAtools 2.4.0. This allowed us to identify outlier samples (1 outlier (1011_0025) in the CSS—BA-NAc neuron transcriptome experiment and 1 outlier (1055_0002) in the BA-NAc neuron tetanus toxin experiment) which were removed prior to further analyses. Using brain cell type-specific marker genes to identify the relative contribution of different cell types to

the RNA sample (mouse visual cortex[70]), the glutamate neuron gene marker *Slca7a7*, as well as the pan-neuronal gene marker *Snap25*, displayed consistent and markedly higher expression than marker genes for GABA (inter)neurons (*Gad1*, *SSt*, *Pvalb*) and each of the glial cell types (astrocyte: *Aqp4*, oligodendrocyte progenitor cell: *Pdgfra*, myelinating oligodendrocyte: *Opalin*, microglia: *Ctss*). Differential gene expression between experimental groups was detected with DESeq2 v1.32.0 using an absolute log2 fold-change of at least 0.5 and a raw *p*-value of ≤0.001. Functional enrichment analysis was conducted with enrichR v3.0 against the mouse-specific pathway collection from KEGG 2019[71].

**Statistical analysis.** Statistical analysis was conducted using Prism (GraphPad, version 9) or SPSS (IBM, version 26). In each experiment, data sets were first assessed for outliers, using the ROUT test in Prism and Boxplot analysis in SPSS; any outliers identified were removed. Next, data were assessed for normal distribution, using the D'Agostino-Pearson normality test in Prism and the Shapiro–Wilk test in SPSS. Subjects that were outliers in terms of behavioural scores in the DRLM and/or REV test were identified in the CSS-behaviour experiment (CSS N = 1) and CSS-fibre photometry-behaviour experiment (CON N = 1, CSS N = 2), and their removal resulted in normally distributed data sets. For *t* tests, homogeneity of variance was assessed using the F test in Prism and Levene's test in SPSS. For ANOVA in Prism, homoscedasticity plots of residuals were checked visually and with Spearman's rank correlation coefficient test, and in SPSS, Levene's test of homogeneity of variance was used. For analysis of the behavioural data for each measure in the DRLM test, 2-way mixed-model ANOVA was applied, with 1 between-subjects factor (e.g. CSS, CON; BA TeTxLC × NAc retro-Cre, BA TeTxLC × NAc retro-mCherry, BA VEH × NAc retro-Cre) and 1 within-subjects factor (test day). For analysis of the behavioural data for each measure in the REV test, either a *t* test or 1-way between-subjects ANOVA was applied. In the BA-NAc neuron tetanus toxin experiment and the BA-NAc neuron DREADDs experiment, 2 control groups were used: in the case of a significant effect of group, planned contrasts were conducted, firstly comparing the 2 control groups only and, if these did not differ significantly from each other, secondly comparing the combined control groups with the experimental group. In the CSS-CON experiments, significant main or interaction effects were analysed *posthoc* using Tukey's or Sidak's multiple comparison test. For analysis of the fibre-photometry data for each phase in the DRLM test, a 3-way mixed-model ANOVA was applied with the between-subjects factor of group and the within-subjects factors of test (test 1 and test 3) and time-normalized interval or time interval. For analysis of the fibre photometry data for each phase in the REV test, a 2-way mixed-model ANOVA was applied with the between-subjects factor of group and the within-subjects factor of time-normalized interval or time interval. In the case of significant main or interaction effects, Tukey's or Sidak's posthoc multiple comparison test was conducted.

Data are reported primarily as mean ± standard error of the mean (SEM) and in supplementary tables as mean ± standard deviation (SD). Statistical significance was set at *p* ≤ 0.05. When a factor had non-significant *p* values for both main effect and interaction effect, *p* values are reported as ≥lower *p*-value.

**Reporting summary.** Further information on research design is available in the Nature Portfolio Reporting Summary linked to this article.

## Data availability

Raw sequencing data and gene expression matrices from the CSS—BA-NAc neuron transcriptome and BA-NAc neuron tetanus toxin transcriptome experiments were deposited in the Gene Expression Omnibus and can be accessed with accession codes GSE216587 and GSE216588, respectively.

## Code availability

The code that was used to process and analyse the expression data is available on https://github.com/platrad-uzh/CSS_effects_on_BA-NAc_rnaseq.

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

## Acknowledgements

This research was funded by the Swiss National Science Foundation (31003A_179381 to C.R.P.) and by a Boehringer-Ingelheim Innocentive grant (Mouse models of apathy and helplessness, to C.R.P.). We are grateful to Björn Henz and Alex Osei for animal caretaking, and to Hannah Wyatt, Susanne Zach and Diana Santacruz for technical support.

## Author contributions

L.M. acquired, analysed and interpreted data and drafted the manuscript; C.I. designed the study, acquired, analysed and interpreted data and drafted the manuscript; G.B. designed the study and acquired, analysed and interpreted data; A.G. acquired, analysed and interpreted data and wrote analysis scripts; G.P. acquired, analysed and interpreted data; N.C-H. acquired, analysed and interpreted data; H.S. acquired data; Y.S. created hardware and software for fibre photometry; J-C.P. constructed and produced viral vectors; K.D.B. designed the study and drafted the manuscript; C.V. acquired and analysed data; F.F-A. analysed and interpreted data; G.A-L. analysed and interpreted data; B.H. designed the study and drafted the manuscript; C.R.P. conceived and designed the study, interpreted the data and drafted the manuscript.

## Competing interests

K.D.B., C.V., F.F-A., G-A-L. and B.H. are employees of Boehringer Ingelheim Pharma GmbH & Co KG. C.R.P. has received research funding from Boehringer Ingelheim Pharma GmbH & Co KG. All other authors report no biomedical financial interests or potential competing interests.
