## [Peer Review File · Communications Biology]

Reviewers' comments:

Reviewer #1 (Remarks to the Author):

In the manuscript, the authors attempt to reveal the association between stress deficits in reward behavior and dysregulation of amygdala-nucleus accumbens pathway function. They found that stress disruption of reward processing involves the BA-NAc neural pathway. However, this is a widely known concept that many papers have already proved. The authors did not go deep enough to explore the underlying mechanism, although they explored the effects of CSS on population-level transcriptome expression of BA-NAc neurons (only descriptive results). The novelty greatly mitigates my enthusiasm for this work to be published in this journal.

Reviewer #2 (Remarks to the Author):

In this manuscript, the author used a previously established behavioral paradigm and circuit specific manipulation and found the possible involvement of BA-NAc circuit in chronic social stress (CSS) (or social defeat) in causing altered reward directed behavior (learning and valuation of reward) in mice. Although some findings of this study seem interesting, there are some major concerns.

Major concerns:

(1) As the first main figure, Fig 1 mainly replicated their previous findings in Azzinnari, D. et al. 2014, despite some minor differences. This part of data does not appear to provide any novelty and strong justification to put as a main figure is necessary.

1. In Fig 2, the author used fibre photometry to record neural activity of NAc projecting BA neurons (BA-NAc neurons) during DRL task. However, considering the calcium sensor/fibre photometry has relatively low spatiotemporal resolution and also reports only the bulk Ca²⁺ activity of NAc-projecting BA neurons, using photometry alone here is hard to provide convincing conclusion as the authors claimed.

2. In Fig 3, the study on effect of CSS on the transcriptome of BA-NAc neurons does not seem to be coherent with the current study as it does not generate any significant nor complementary findings, but rather is limited by the heterogeneity in BA neuron population, and/or the potential opposing effects of CSS on the transcriptions of these neurons.

3. In Fig 4, 5 chronic manipulation (TeTxLC inhibition/DREADDs activation) of BA-NAc neurons could only capture very limited aspects of CSS-induced reward-related behavioral deficits and is therefore not very convincing. The involvement of NAc-projecting amygdala neurons in chronic stress induced maladaptive reward and motivation behaviors have been previously mentioned, therefore diminished the novelty and significance of this piece of study (Dieterich et al., 2021, <https://doi.org/10.3389/fnbeh.2021.643272>)

4. Although the author has given some interpretations and proposed some hypotheses (e.g. in Fig 6), the evidence based on the current experiments was not sufficient to support their arguments.

Minor concerns:

1. The writing is difficult to follow. The main text contains too much redundant/unnecessary information. This is reflected in both content (data) and narrative.

2. The mice in Fig 2 Control group seem to have higher motivation to approach the food pellet than the Control group in Fig 1, shown as lower DS and ITI response latency (Fig 1 e,f and Fig 2d,e.). This is also the case for CSS group in Fig 1 and Fig 2. The author did not explain this discrepancy.

3. The formats of some figures are not consistent. For example, the interval between ticks of y-axis in Fig 1d and Fig 2c are not consistent

4. The TeTxLC and DREADS experiments in Fig 4 and Fig5 require control experiments to exclude the possible effects of drugs on the basal locomotive ability of mice.

Reviewer #3 (Remarks to the Author):

In their current study, Madur and collaborators investigate the impact of chronic stress on reward processes and associated neuronal correlates. Using an elegant combination of behavioral models for stress vulnerability, physiological recordings, and chemogenetic approaches, the authors established that stress disruption of reward processing impacts the BA-NAc neural circuit functions. The authors observed opposite activity changes in reward (learning) neurons and aversion (effort) neurons in the BA-NAc pathway following chronic stress associated with the observed reward processing alterations following chronic stress.

The question asked by the authors is intriguing and timely. The results are very interesting and thus may provide useful information to the field of neural circuit activity and stress-related neuropsychiatric conditions. While the overall experimental strategy is well designed and executed, the current dataset requires a few supplemental analyses to exclude potential hypotheses that would ultimately expand the scope of the study. Additional technical details and text corrections should be performed to help interpret the current data set. My current enthusiasm for the manuscript would increase even more if the authors addressed the outlined points section below as much as possible.

Major Comments:

1- The authors observed that mice exposed to a chronic social stress paradigm develop altered reward learning and efforts. These results are very interesting as they define the stress impact on reward processing and learning. Several studies have shown that a chronic social stress paradigm induces individual behavioral and neuronal phenotypes. To increase the impact of the current studies and link the current results to the existing literature, the authors should identify if the alterations of reward processing and efforts are similar across the stressed group or if they correlate with social behavior alterations, anxiety, or stress hormones responses.

2- The authors performed elegant fiber photometry recordings of BA-NAc pathway neurons during assessments of reward learning and efforts. The authors should justify why they analyzed trials 1-20 and then specifically trials 11-20. The authors should also analyze and assess the changes in activity dynamics across each trial independently. In complementary analyses, the authors should perform the same signal analyses on DS trials without responses. The authors need to provide the analyses of BA-NAc activity during PR5-9-13 in CSS mice, similar to Figure 2m in the main results section. The authors should further clarify how the photometry data set was offset to zero for each z-score, trial, and mouse, in Figure 2h, p, q.

3- The authors observed that tetanus-toxin-light-chain manipulation induced behavioral alterations that were similar to the stress-induced behavioral alterations of reward processing and efforts. Yet, to fully support their hypothesis, the authors should aim to reverse the stress-induced neuronal function

alterations by activating the BA-NAc pathway. This behavioral rescue would confirm that stress-induced decreased BA-NAc activity results in altered reward processing and effort, and also significantly increase the impact of the current manuscript.

4- The authors performed DREADDs activation of BA-NAc neurons on reward-directed behavior. Interestingly they observed that tetanus-toxin-light-chain manipulation and DREADDs activation of BA-NAc neurons impaired REV #choice pellets to the same extent. The authors should address these interesting opposite results. In particular, are BA-NAc different neuronal populations targeted by tetanus-toxin-light-chain or AAV? The authors should perform immunohistochemistry investigations and characterize the nature of the manipulated neurons. The authors need to use inhibitory DREADDs and manipulate BA-NAc neurons activity during reward processing and efforts measurement to strengthen the current data set and interpretation.

Minor Comments:

- The authors should describe the analysis performed to determine the sample size.
- The authors should describe the meaning of "ab, abc" etc.. presented in figure 2h,p.
- The authors need to provide quantification analyses of BA-NAc GCaMP expression levels in CSS and control mice.

Responses to Reviewers

Reviewers' comments:

Reviewer #1 (Remarks to the Author):

In the manuscript, the authors attempt to reveal the association between stress deficits in reward behavior and dysregulation of amygdala-nucleus accumbens pathway function. They found that stress disruption of reward processing involves the BA-NAc neural pathway. However, this is a widely known concept that many papers have already proved. The authors did not go deep enough to explore the underlying mechanism, although they explored the effects of CSS on population-level transcriptome expression of BA-NAc neurons (only descriptive results). The novelty greatly mitigates my enthusiasm for this work to be published in this journal.

To the very best of our knowledge, based on regular and extensive searches, there have been no previous published studies of either (1) the effects of (acute or) chronic stress on reward-directed behaviour and concurrent activity/status of BA-NAc neurons, or; (2) the effects of chronic manipulations of BA-NAc neurons that aim to mimic the effects of chronic stress on these neurons, on reward processing. There is a single publication (Dieterich et al., 2021), referred to by Reviewer 2, that describes the effects of chronic corticosterone administration (increased corticosterone is one of the many physiological changes induced by stress) on reward processing, and that acute activation of BA-NAc neurons reduces these effects. Our iterative study differs markedly from this single previous study of this subject, in its aims, design, methods, findings, and the conclusions that can be drawn from it. In short, our study/manuscript would be the first publication on this subject, to the very best of our knowledge.

Reviewer #2 (Remarks to the Author):

In this manuscript, the author used a previously established behavioral paradigm and circuit specific manipulation and found the possible involvement of BA-NAc circuit in chronic social stress (CSS) (or social defeat) in causing altered reward directed behavior (learning and valuation of reward) in mice. Although some findings of this study seem interesting, there are some major concerns.

Major concerns:

(1) As the first main figure, Fig 1 mainly replicated their previous findings in Azzinnari, D. et al. 2014, despite some minor differences. This part of data does not appear to provide any novelty and strong justification to put as a main figure is necessary.

We are of the opinion that it is very important to retain Fig.1 as a main figure even though the data presented for the model of chronic social stress-induced reduced reward interest/learning and reward-to-effort valuation are a replication of the original publication describing the model (not Azzinnari et al., 2014, but Kukul'ova et al., 2018). Animal models that combine chronic stress manipulations with a behavioural readout of reward-directed conditioned behaviour are extremely rare; the majority of research in this area is based on the unconditioned behavioural readout of the sucrose preference test (see Willner, 2017. *Neurobiol. Stress*, 6, 78-93, doi: 10.1016/j.ynstr.2016.08.002). Therefore, most readers (neuroscientists, psychologists, psychiatrists) will not be familiar with the chronic stress-induced reduced reward learning and effortful motivation model and the conditioned behavioural tests it uses. Furthermore, we would also like to point out that replication of behavioural effects in sophisticated translational rodent models is not to be taken for granted, and that the current model provides a rare example of a robust and reproducible model

of chronic stress-induced deficits in conditioned reward behaviour (please also see response to Minor concern 2 in this regard).

1. In Fig 2, the author used fibre photometry to record neural activity of NAc projecting BA neurons (BA-NAc neurons) during DRL task. However, considering the calcium sensor/fibre photometry has relatively low spatiotemporal resolution and also reports only the bulk Ca²⁺ activity of NAc-projecting BA neurons, using photometry alone here is hard to provide convincing conclusion as the authors claimed.

With the appropriate combination of viral vectors, calcium-sensor fibre photometry can be used to measure the activity of neuronal populations (1) in specific pathways and therefore with high spatial resolution, and (2) in real time, such that activity can be correlated with on-going sequences of behaviour at a resolution of 50 ms or less. Whilst this is not comparable to single cell electrophysiological recording, the pathway-specific fibre photometry method that we have used is the very same as that used in a substantial number of studies reporting on the involvement of specific neural pathways in specific behavioural processes (one example among many during the last 8 years : Gunaydin et al., 2014, Cell, 157, 1535-1551, doi: 10.1016/j.cell.2014.05.017). Our pathway-specific calcium-activity fibre-photometry study of a population of BA-NAc glutamate neurons yielded important and novel evidence that: (1) The activity of BA-NAc neurons is altered during specific reward processes; namely it increases during reward learning and also increases in line with the effort required to obtain reward. (2) Chronic social stress impacts on these BA-NAc neuron activity changes; namely it inhibits the increase during attenuated reward learning and accentuates the increase during attenuated effortful responding for reward. Follow up BA-NAc neural pathway-specific experiments then demonstrated that chronic inhibition (TeTxLC) of BA-NAc neurons inhibits reward learning and chronic activation (DREADDs) of BA-NAc neurons inhibits effortful responding for reward. Parsimonious interpretation of these iterative findings suggests that this neuronal population actually comprises two substantial sub-populations, namely one of reward neurons activated by reward and involved in reward learning (discriminative reward learning memory test), and one of aversion neurons activated by effortful demands (reward-to-effort valuation test). In the manuscript, we are very clear that these findings (1) constitute important, novel information, and (2) will require subpopulation-specific experiments to investigate the hypotheses generated by the findings. Rather than being an inappropriate method, sub-population-specific (i.e. reward neurons only, aversion neurons only) BA-NAc calcium-sensor fibre photometry will be a highly appropriate method to test the subpopulation-specific hypotheses generated by the current study. Such experiments are beyond the scope of the current iterative study/manuscript.

2. In Fig 3, the study on effect of CSS on the transcriptome of BA-NAc neurons does not seem to be coherent with the current study as it does not generate any significant nor complementary findings, but rather is limited by the heterogeneity in BA neuron population, and/or the potential opposing effects of CSS on the transcriptions of these neurons.

The data presented in Fig.3 are absolutely coherent with the current study. They depict the effects of chronic social stress on the transcriptome expression status of the population of BA-NAc glutamate neurons investigated in the other, iterative experiments. To the best of our knowledge, this is the first report of chronic stress effects on this population of neurons, and the experimental procedure was validated carefully to ensure that the tissue collected by fluorescence-based laser capture microdissection was indeed obtained from BA-NAc glutamate neurons (500 neurons per sample). The main finding is that CSS leads to an increase in the expression of a moderate number of genes, and these genes are enriched in pathways related to neuronal signalling. Taking the evidence obtained in the CSS-fibre photometry, TeTxLC and DREADDs experiments, and integrating this with evidence that the BA-NAc neuron population studied comprises reward neurons and aversion neurons, we were able to conclude in the Discussion: "Therefore, the main finding that CSS resulted in dysregulated expression of only a moderate number of genes could be due to the heterogeneous neuron population and to CSS exerting specific or even opposing effects on the transcriptomes of

BA-NAc reward neurons and aversion neurons.” That is, the various, iterative experiments of the present study provide extensive new evidence for the effects of chronic stress on BA-NAc neurons that is both of major importance per se and will inform future studies at the neuronal sub-population level.

It is also important to emphasize here that, when we began this extensive study, the available data indicated that the majority of BA-NAc neurons were monovalent reward responsive neurons and only a minority were aversion responsive (Kim et al., 2016, *Nature Neurosci.*, 19, 1636-1646, 10.1038/nn.4414; Beyeler et al., 2018, *Cell Reports*, 22, 905-918, 10.1016/j.celrep.2017.12.097). However, in a set of experiments that we have conducted in parallel with this study, we have obtained evidence that these BA-NAc subpopulations of monovalent reward and aversion neurons are of similar size. To clarify this specific but relevant point we have added the following text to the Discussion:

The current novel findings for the inter-relationships between chronic social stress, activity of BA-NAc glutamate neurons, and specific aspects of reward processing and behaviour, and a hypothesis to account for causal mechanisms, are summarised in Fig. 6. The current findings are consistent with there being two major subpopulations of BA-NAc neurons, one of monovalent neurons activated by reward and one of monovalent neurons activated by aversion including effort. Although previous studies have reported that there are more such reward neurons than aversion neurons (Beyeler et al., 2016, 2018; Kim et al., 2016), a recent study from our laboratory indicates that, for the intermediate BA-NAc at least, the two neuronal subpopulations are of similar size (Poggi et al., submitted).

3. In Fig 4, 5 chronic manipulation (TeTxLC inhibition/DREADDs activation) of BA-NAc neurons could only capture very limited aspects of CSS-induced reward-related behavioral deficits and is therefore not very convincing. The involvement of NAc-projecting amygdala neurons in chronic stress induced maladaptive reward and motivation behaviors have been previously mentioned, therefore diminished the novelty and significance of this piece of study (Dieterich et al., 2021, <https://doi.org/10.3389/fnbeh.2021.643272>)

Rather than “only captur(ing) very limited aspects of CSS-induced reward-related behavioral deficits”, the chronic manipulations of TeTxLC inhibition of BA-NAc neurons and DREADDs activation of BA-NAc neurons actually replicated specific and different aspects of CSS-induced reward-related behavioral deficits. In summary, the iterative experiments conducted provide novel evidence that (1) chronic social stress-induced deficits in reward learning and reward-to-effort motivation co-occur, respectively, with decreased and increased activity of a BA-NAc neuronal population; and (2) these effects can be recapitulated by chronic inhibition (TeTxLC) or chronic activation (DREADDs) of this neuronal population, respectively. These reward learning-specific effects of chronic TeTxLC BA-NAc inhibition and reward-to-effort valuation-specific effects of DREADDs BA-NAc activation, each replicating one specific aspect of the CSS effect, are novel and cardinal findings of this study.

Whilst the cited paper by Dieterich et al. (2021) is very interesting, the parameters investigated are fundamentally different from those in our study. Rather than being exposed to chronic social stress, mice were exposed to chronic corticosterone (CORT) administration. Whilst increased CORT is one component of the stress response that does have effects on brain state, it is fundamentally different to exposure to an uncontrollable and unpredictable psychological stressor which activates many brain systems, just one component of which is activation of the HPA axis and increased CORT synthesis and release. With regard to their translational relevance (aetiological validity) to the inter-relationships between human psychosocial stress, BA-NAc pathway function and dysfunctional reward processing, CSS is of higher relevance than chronic CORT exposure. There is another fundamental difference between the two studies in that Dieterich et al. used acute DREADDs activation of the BA-NAc pathway to investigate its effects on reward behaviour in mice

exposed to chronic CORT, whereas we used chronic DREADDs activation to investigate whether this would replicate specific effects of CSS on reward behaviour. Given the fundamental differences between the Dieterich et al. study and our study, we did not consider it essential to refer to it in our manuscript. However, to acknowledge that Dieterich et al. (2021) is nonetheless the published study closest to our study in terms of overall aims, we have now added the following text to the Discussion:

“In a previous study of some relevance here (Dieterich et al., 2021), when mice were administered corticosterone (CORT) chronically they did not show a decrease in effortful reward responding but did make fewer feeder responses on a low-effort schedule, and DREADDs activation of BA-NAc neurons directly before low-effort behavioural testing increased feeder responses in both chronic-CORT and control mice; the respective findings of the two studies highlight the importance of the manipulation used and the duration of DREADDs activation of the BA-NAc neurons.”

4. *Although the author has given some interpretations and proposed some hypotheses (e.g. in Fig 6), the evidence based on the current experiments was not sufficient to support their arguments.*

The iterative experiments conducted provide novel evidence that (1) chronic social stress-induced deficits in reward learning and reward-to-effort motivation co-occur, respectively, with decreased and increased activity of a BA-NAc neuronal population; and (2) these effects can be recapitulated by chronic inhibition or chronic activation of this neuronal population, respectively. To account for these novel and important findings, we propose that CSS might be acting to inhibit BA-NAc reward neurons and activate BA-NAc aversion neurons. This novel hypothesis is based on the novel findings presented. We agree with the Reviewer that in the Discussion text relating to Fig. 6 and the legend for Fig. 6, we have inter-mingled somewhat the interpretation of current findings and the hypotheses based on these findings. We have revised the text accordingly. As the main example of this, we have changed the figure legend from:

Figure 6. Interpretation of current findings for the inter-relationships between chronic social stress, activity of BA-NAc neural pathway activity and reward processing, and integration with existing evidence.

to:

“Figure 6. Summary of current findings for the inter-relationships between chronic social stress, activity of BA-NAc neural pathway activity and reward processing, and a proposed hypothesis for causal mechanisms.” We have also specified which part of the legend refers to findings and which part to hypothesis.

Minor concerns:

1. *The writing is difficult to follow. The main text contains too much redundant/unnecessary information. This is reflected in both content (data) and narrative.*

The senior author is a native English speaker with extensive experience in scientific writing, and it is uncommon to receive the feedback that the writing is difficult to follow. Because we have used a large number of different methods, and scientists from different backgrounds (e.g. neuroscience, psychology, psychiatry) are likely to be interested in the paper, we are of the opinion that it is important to provide a summary of the methods – the only text that could be deemed redundant - in the Results section, in addition to the detailed information provided in the Methods section. We have gone through the text carefully and divided each of a small number of relatively long sentences into two sentences.

2. *The mice in Fig 2 Control group seem to have higher motivation to approach the food pellet than the Control group in Fig 1, shown as lower DS and ITI response latency (Fig 1 e,f and Fig 2d,e.). This is also the case for CSS group in Fig 1 and Fig 2. The author did not explain this discrepancy.*

It is important to note that the data reported on in Fig. 2 are for mice that have undergone stereotactic surgery for BA and NAc injection of viral vectors and are being tested with a patch cord attached to a cranial optic fibre. Conversely, the data reported for Fig. 1 are for mice that have not undergone these procedures. Furthermore, the two experiments used different test apparatus, with that used for the data in Fig. 2 being adapted for fibre photometry. Therefore, it is not surprising that there are some differences in the behavioural scores across the two experiments. However, the major, consistent finding from the two experiments is that, even under these different test conditions, the model of CSS-induced reductions in DRLM and REV are reproducible and robust. This is not to be taken for granted in such a sophisticated translational model. We have added the text:

“When comparing the two CSS experiments, there were some differences in the absolute behavioural scores in the case of CON mice and CSS mice, which are likely attributable to the various effects of the procedures specific to the fibre photometry experiment (e.g. optic fibre implantation, patch cord attachment, modification of reward dispenser). However, the major, consistent finding from the two experiments is that, even under these different test conditions, the model of CSS-induced reductions in DRLM and REV is reproducible and robust.”

3. The formats of some figures are not consistent. For example, the interval between ticks of y-axis in Fig 1d and Fig 2c are not consistent

Thank you. This inconsistency has now been corrected.

4. The TeTxLC and DREADS experiments in Fig 4 and Fig5 require control experiments to exclude the possible effects of drugs on the basal locomotive ability of mice.

The Reviewer is correct, and this is why we conducted such control experiments and reported them in the original manuscript.

For the TeTxLC experiment, supplementary Fig. 4c shows clearly that there was no effect of any of the viral vector combinations used on basal locomotor behaviour (no drug was used in this experiment). We have now added this information to the relevant section of the Results.

For the DREADDs experiment, supplementary Fig. 7b shows clearly that there was no effect of any of the viral vector combinations used on basal locomotor behaviour, and supplementary Fig. 7g shows clearly that there was no effect of the clozapine study dose of 0.3 mg/kg i.p. on basal locomotor behaviour, and supplementary Fig. 7j shows clearly that there was no effect of clozapine-N-oxide (up to 2 mg/kg i.p.) on basal locomotor behaviour. We have now added this information to the relevant section of the Results.

Reviewer #3 (Remarks to the Author):

In their current study, Madur and collaborators investigate the impact of chronic stress on reward processes and associated neuronal correlates. Using an elegant combination of behavioral models for stress vulnerability, physiological recordings, and chemogenetic approaches, the authors established that stress disruption of reward processing impacts the BA-NAc neural circuit functions. The authors observed opposite activity changes in reward (learning) neurons and aversion (effort) neurons in the BA-NAc pathway following chronic stress associated with the observed reward processing alterations following chronic stress.

The question asked by the authors is intriguing and timely. The results are very interesting and thus may provide useful information to the field of neural circuit activity and stress-related neuropsychiatric conditions. While the overall experimental strategy is well designed and executed,

the current dataset requires a few supplemental analyses to exclude potential hypotheses that would ultimately expand the scope of the study. Additional technical details and text corrections should be performed to help interpret the current data set. My current enthusiasm for the manuscript would increase even more if the authors addressed the outlined points section below as much as possible.

Major Comments:

1- The authors observed that mice exposed to a chronic social stress paradigm develop altered reward learning and efforts. These results are very interesting as they define the stress impact on reward processing and learning. Several studies have shown that a chronic social stress paradigm induces individual behavioral and neuronal phenotypes. To increase the impact of the current studies and link the current results to the existing literature, the authors should identify if the alterations of reward processing and efforts are similar across the stressed group or if they correlate with social behavior alterations, anxiety, or stress hormones responses.

This important point refers to the inter-individual differences in responses to chronic social stress procedures that have been reported by some studies/laboratories when BL/6 mice are tested in terms of whether they show social passive avoidance to a mouse of the same strain (CD-1) and size as the mice used to induce social stress. Relative to controls, the majority (around 65%) of mice show social passive avoidance and are said to be susceptible, and a minority (around 35%) of mice do not show social passive avoidance and are said to be resilient. It is well-established that mice show marked spontaneous inter-individual differences in behaviour, including in passive avoidance. Indeed, a recent study demonstrated that inter-individual differences in passive avoidance before chronic social stress positively predict inter-individual differences in passive avoidance after chronic social stress i.e. trait effects and chronic social stress effects are additive (Milic et al., 2021, *Neurobiol. Stress*, 14, 100290, doi: [org/10.1016/j.ynstr.2020.100290](https://doi.org/10.1016/j.ynstr.2020.100290)). Rather than allocating mice to subgroups based on their passive avoidance trait-state, our approach is to investigate CSS effects in the entire sample of mice, so that trait differences are controlled for between the CSS and CON groups. As can be seen for the behavioural data and the fibre photometry data in Figures 1 and 2, the inter-individual variability is very similar between CSS and control groups for nearly all measures, indicating that, for the reward processes investigated, there is no evidence for subgroups of CSS susceptible and resilient mice. Indeed, the same applies when the behavioural readout is Pavlovian aversion learning and memory (e.g. Just et al., 2018, *PLoS One*, 13, e0191225, doi: [10.1371/journal.pone.0191225](https://doi.org/10.1371/journal.pone.0191225); Adamczyk et al., 2022, *Biol. Psychiatry: GOS*, 2, 470-479, doi: [10.1016/j.bpsgos.2021.11.006](https://doi.org/10.1016/j.bpsgos.2021.11.006)).

To bridge this difference in approach between different studies/models, we have added the following text:

To the Results:

In this study, the mean duration of daily attack experienced by CSS mice was 45–50 s; all CSS mice displayed submissive behaviour and vocalization during the proximal stressor.

To the relevant section of the Discussion:

These CSS effects on non-social behaviour are obtained without subgrouping mice according to whether or not they passively avoid CD-1 male mice subsequent to the CSS period, a procedure that is commonly used in other studies (e.g. Golden et al., 2011; Krishnan et al., 2007). It is well-established that mice show marked spontaneous inter-individual differences (traits) in behaviour, including in passive avoidance; indeed, a recent study demonstrated that inter-individual differences in passive avoidance before chronic social stress positively predict inter-individual differences in passive avoidance after chronic social stress i.e. trait effects and chronic social stress effects are additive (Milic et al., 2021). Rather than allocating mice to subgroups based on their passive

avoidance trait-state, our approach is to investigate CSS effects in the entire sample of mice, so that trait differences are controlled for between the CSS and CON groups. As can be seen for behavioural data and fibre photometry data (Figures 1 and 2), inter-individual variability is comparable between CSS and CON groups for nearly all measures, indicating that, for the reward processes investigated, there is no evidence for subgroups of CSS mice. Indeed, the same applies when the behavioural readout is Pavlovian aversion learning and memory (e.g. Just et al., 2018; Adamczyk et al., 2021). As described in Results, all CSS mice displayed submissive behaviour during proximal attack sessions. Using this experimental design, sample sizes of 12 are sufficient to yield significant CSS effects without subgrouping.

2- The authors performed elegant fiber photometry recordings of BA-NAc pathway neurons during assessments of reward learning and efforts. The authors should justify why they analyzed trials 1-20 and then specifically trials 11-20. The authors should also analyze and assess the changes in activity dynamics across each trial independently. In complementary analyses, the authors should perform the same signal analyses on DS trials without responses. The authors need to provide the analyses of BA-NAc activity during PR5-9-13 in CSS mice, similar to Figure 2m in the main results section. The authors should further clarify how the photometry data set was offset to zero for each z-score, trial, and mouse, in Figure 2h, p, q.

Thank you for this comment and the recommendations for additional data presentation.

As stated in the original manuscript, in the discriminative reward learning-memory (DRLM) test, the reason for analysing trials 1-20 was as follows:

“Each test comprised 40 trials: across the 3 tests, the number of trials with DS feeder response was lower in CSS mice (18.4±5.6 per test, mean±SD) than CON mice (24.4±4.4 per test) (p = 0.01, supplementary Table S2). Given that average DS response latency was lowest in trials 1-20 and then began to increase, as did the number of trials without a response, data analysis was conducted for trials 1-20.”

As stated in the original manuscript, the reason for conducting an additional analysis for DRLM test trials 11-20 was as follows:

“To increase sensitivity to detecting effects attributable to within-session learning, analysis was restricted to trials 11-20 with a DS response”

We have now added extensive supplementary data to further justify and clarify the analysis of the DRLM test data undertaken in the fibre photometry experiment:

In the Results, we have added the analysis for trial 1-10 specifically, as new supplementary Figure 2a, demonstrating the absence of a CSS effect in these trials, in contrast to trials 11-20:

“In trials 1-10 with a DS feeder response, there was no CSS effect (supplementary Fig. 2a).”

To provide further information on how BA-NAc neuronal population activity developed within a test session, for DRLM test 3, we have now added the fibre photometry data for CON and CSS mice for trials 1, 5, 10, 15 and 20 independently. For each of these trials, we provide the mean and SEM values for those mice that made a DS response at this trial. These additional data are given as new supplementary Fig. 2b, and the following text has been added to the Results:

“The within-session divergence of BA-NAc neural Ca²⁺ activity in CSS and CON mice can also be visualised when the data are presented for specific trials: supplementary Fig. 2b gives the Ca²⁺ activity data, for those mice which made a DS feeder response, for trials 1, 5, 10, 15 and 20 separately.”

The complementary analysis of DS trials without feeder responses has now been added. The additional data are given as **new supplementary Fig. 2c** and the following text has been added to the Results:

“The CSS effect on BA-NAc neural Ca²⁺ activity was specific to trials in which mice made a DS feeder response, with there being no significant difference between CSS and CON mice in trials without a DS feeder response (supplementary Fig. 2c).”

In the REV test, the analysis of BA-NAc activity during PR 9 and 13 for the same phase as shown in Fig 2m, has now been added. **New supplementary Fig. 3b** provides a direct CSS vs CON comparison at PR 9 and PR 13; the main CSS vs CON comparison, at PR 5, is already given in Fig. 2o. Further to the additional graphs the following text has been added to the Results:

“(It is noteworthy that mean BA-NAc neural activity during the post-operand phase was also higher in CSS than CON mice at PR 9 and PR 13 (supplementary Fig. 3b) but due to the low sample size for CSS mice at these higher PRs statistical analysis was not conducted.)”

The information on how the photometry data set was offset to zero in Fig. 2h, 2o and 2p, i.e. at each phase of the DRLM test and REV test analysis, had been added to the Results:

Text for DRLM test:

“As for the DS-on phase, trial-specific mean Ca²⁺ activity during the 10 s prior to DS onset provided the measure of baseline activity.”

Text for REV test:

“Trial-specific mean BA-NAc activity during 10 s prior to the onset of operand responding constituted baseline activity, and was used for each phase of analysis.”

3- The authors observed that tetanus-toxin-light-chain manipulation induced behavioral alterations that were similar to the stress-induced behavioral alterations of reward processing and efforts. Yet, to fully support their hypothesis, the authors should aim to reverse the stress-induced neuronal function alterations by activating the BA-NAc pathway. This behavioral rescue would confirm that stress-induced decreased BA-NAc activity results in altered reward processing and effort, and also significantly increase the impact of the current manuscript.

The chronic expression of tetanus toxin light chain (TeTxLC), a well-established method for inhibition of glutamate release i.e. neuron inhibition, in BA-NAc neurons did indeed replicate the inhibitory effect of CSS on discriminative reward learning and had a partial inhibitory effect on reward-to-effort valuation. In contrast to the proposal of the Reviewer, and as reported in detail in the manuscript, when the BA-NAc pathway was activated chronically using excitatory DREADDs, there was no effect on discriminative reward learning but a reduction in motivation, similar to the motivational effects of CSS, in both the discriminative reward learning test and the reward-to-effort valuation test. Therefore, we hope that the Reviewer would agree that there is no rational basis for the hypothesis that BA-NAc neural pathway activation will rescue the CSS-induced behavioural deficits in reward processing. As we put forward in the Discussion (and Fig. 6), one model that can explain our findings is that the population of BA-NAc neurons that we have studied comprises two subpopulations, namely reward responsive neurons and aversion responsive neurons. CSS leads to reduced reward learning associated with decreased BA-NAc neuron activity, and chronic TeTxLC BA-NAc neuron inhibition replicates this effect; we propose that this is primarily mediated by the inhibited BA-NAc

reward neurons. CSS leads to reduced reward-to-effort valuation associated with increased BA-NAc neuron activity, and chronic DREADDs BA-NAc neuron activation replicates this effect; we propose that this is primarily mediated by the activated BA-NAc aversion neurons due to the effortful demands of the task. Having conducted this series of experiments that yielded the novel data that can be accounted for by this model, we conclude that the next step will be to conduct subpopulation-specific studies of the reward neurons and aversion neurons to test the hypotheses generated. Such next step experiments are beyond the scope of the current extensive study/manuscript.

It is also important to emphasize here that, when we began this study, the available data indicated that the majority of BA-NAc neurons were monovalent reward responsive neurons and only a minority were aversion responsive (Kim et al., 2016, *Nature Neurosci.*, 19, 1636-1646, 10.1038/nn.4414; Beyeler et al., 2018, *Cell Reports*, 22, 905-918, 10.1016/j.celrep.2017.12.097). However, in a set of experiments that we have conducted in parallel with this study, we have obtained evidence that these BA-NAc subpopulations of monovalent reward and aversion neurons are of similar size. To clarify this specific but relevant point we have added the following text to the Discussion:

The current novel findings for the inter-relationships between chronic social stress, activity of BA-NAc glutamate neurons, and specific aspects of reward processing and behaviour, and a hypothesis to account for causal mechanisms, are summarised in Fig. 6. The current findings are consistent with there being two major subpopulations of BA-NAc neurons, one of monovalent neurons activated by reward and one of monovalent neurons activated by aversion including effort. Although previous studies have reported that there are more such reward neurons than aversion neurons (Beyeler et al., 2016, 2018; Kim et al., 2016), a recent study from our laboratory indicates that, for the intermediate BA-NAc at least, the two neuronal subpopulations are of similar size (Poggi et al., submitted).

4- The authors performed DREADDs activation of BA-NAc neurons on reward-directed behavior. Interestingly they observed that tetanus-toxin-light-chain manipulation and DREADDs activation of BA-NAc neurons impaired REV #choice pellets to the same extent. The authors should address these interesting opposite results. In particular, are BA-NAc different neuronal populations targeted by tetanus-toxin-light-chain or AAV? The authors should perform immunohistochemistry investigations and characterize the nature of the manipulated neurons. The authors need to use inhibitory DREADDs and manipulate BA-NAc neurons activity during reward processing and efforts measurement to strengthen the current data set and interpretation.

CSS led to reduced reward-to-effort valuation (REV) associated with increased BA-NAc neuron activity. Although both chronic TeTxLC inhibition and chronic DREADDs activation of BA-NAc neurons led to inhibition of REV, the effect of DREADDs activation was greater. As put forward in the Discussion, including the model in Fig. 6, we propose that reduced REV is primarily mediated by the activation of BA-NAc aversion neurons by effortful demand. As explained in the Discussion and model: It has been reported that one effect of BA aversion neuron activation is inhibition of BA reward neurons (Kim et al. 2016, *Nat. Neurosci.*, 19, 1636-1646, doi: 10.1038/nn.4414): the mild effect of TeTxLC inhibition on REV could therefore be the net outcome of inhibition of reward and aversion neurons. As explained in the response to point 3, having conducted this series of experiments that yielded the novel data that support this model, we conclude that the next step will be to conduct subpopulation-specific studies of the reward neurons and aversion neurons to test the hypotheses generated. Such next step experiments are beyond the scope of the current study/manuscript.

With regard to the Reviewer's question/hypothesis "are BA-NAc different neuronal populations targeted by tetanus-toxin-light-chain or AAV (we assume DREADDs is meant here)?" we are confident that the TeTxLC and DREADDs AAV vector systems are targeting the same neuron population because in both cases the same retrograde Cre-vector is injected into the same region of the NAc, and the FLEX-TeTxLc (inhibition) and FLEX-DREADDs (activation) vectors are injected into the same region of the BA. In the Results, we have now included an additional supplementary figure to allow direct visualisation of the similar distribution of expression of the two vectors by BA-Nac neurons, and added the corresponding text to the Results:

Representative images showing the coronal distribution of TeTxLC-targeted BA-NAc neurons in the BA are given in supplementary Fig. 9.

Representative images showing the coronal distribution of DREADDs-targeted BA-NAc neurons in the BA are given in supplementary Fig. 9; the distribution and density are similar to those obtained in the TeTxLC BA-NAc neuron inhibition experiment.

Concerning the issue of reduced reward-to-effort valuation coincident with increased BA-NAc neural activity in CSS mice, we have cited a recent human depression study that gives this finding further translational relevance, in the Discussion:

This novel finding for increased BA-NAc neural activity associated with effort in CSS mice is relevant to the human evidence that sensitivity to effort cost is increased (i.e. REV is reduced) in MDD (Vinckier et al., 2022).

Based on the above information, we hope that the Reviewer would agree that there is no rational basis for conducting an experiment with inhibitory DREADDs, because we have already conducted an experiment with inhibitory TeTxLC. As explained above in responses to points 3 and 4, we hypothesize that furthering understanding of the results obtained in this study will necessitate separate targeting of the subpopulations of BA-NAc reward neurons and aversion neurons in a follow-up study, using CSS and subpopulation-specific fibre photometry, and subpopulation-specific chronic TeTxLC and DREADDs viral vector systems. We conclude the Discussion of the manuscript with this statement.

Reviewer 3, Minor concerns

- *The authors should describe the analysis performed to determine the sample size.*

We have added the following information to Methods/Animals:

Experiments with the chronic social stress (CSS)-induced attenuated reward processing model that provides the basis for the current study are typically conducted with n=12-14 CSS versus n=12-14 control mice. This sample size yields robust and reproducible CSS effects (e.g. Adamczyk et al., 2022; Kukulova et al., 2018; Münster et al., 2022), and was therefore the sample size used for behavioural experiments in the current study. Because we did not have prior data for the effects of CSS on BA-NAc neural activity during reward processing, for the fibre photometry experiment we aimed for a sample size of 18-20 mice per group.

- *The authors should describe the meaning of "ab, abc" etc.. presented in figure 2h,p.*

We have added the following information to each figure legend for which it is relevant:

Test days and intervals indicated by different letters were significantly different in Tukey's multiple comparisons test: e.g. a vs b, a vs c, b vs c, $p < 0.05$ or lower.

- *The authors need to provide quantification analyses of BA-NAc GCaMP expression levels in CSS and control mice.*

In addition to the representative microscope images of Nissl-stained coronal brain sections showing the location of the optic fibre implant/tip in the BA and the colocalized AAV vector-expressed GCaMP6 fluorescence (supplementary Fig. 1a), we have now added a supplementary figure 1b indicating the estimated location of the optic fibre tip and GCaMP expression in the BA for each mouse included in the data analysis of the fibre photometry experiment. The following text has been added to the Methods and the figure is referred to in the Results:

“and supplementary Fig. 1b provides the estimated BA location of the optic fibre tip and GCaMP6 expression in CON and CSS mice, based on histological assessments.”

As is typical for such experiments across laboratories, we do not have an ex vivo method for quantifying GCaMP6 expression levels. Furthermore, also in line with general opinion, for the following reasons we are of the opinion that there is no need to do so. (1) It is the case that only mice that had a reliably detectable GCaMP6 signal during fibre photometry were included in the experiment (see Methods/Effects of CSS on BA-NAc glutamate neuron activity during reward-directed behaviour using fibre photometry). (2) Because fibre photometry signal intensity was z-scored (normalized) to baseline for each mouse and each trial, any inter-individual differences in absolute signal intensity will not contribute to inter-individual differences in the data used for statistical analysis.

Reviewers' comments:

Reviewer #2 (Remarks to the Author):

I appreciate the efforts that the authors try to address the concerns being raised. However, the major concern is still the novelty of this study. I don't think there is real conceptual advancement on the BA-NAc pathway involving stress-induced impairment of reward processing, although the authors emphasized chronic social stress here. Just as Reviewer 1 mentioned, the current study does not go deep enough to explore the underlying mechanism even in the revised version, such as how the specific reward and aversive neurons act and interact in CSS-induced impairment in reward processing. And I still doubt the necessity of retaining Fig 1, especially because of the fact is that Fig 2 has included all the messages shown in Fig 1. Application of transcriptome expression appears to be an interesting part of the study, but again there is no extended exploration therefore does not provide significant novelty. This result also appears to be dissociated from other parts.

Reviewer #3 (Remarks to the Author):

In the revised manuscript, Madur and collaborators have adequately revised the manuscript and addressed important questions regarding the interpretation of the analyses and results. Overall, the study, in its revised form, provides information to the research field of stress-related behavioral disorders and neuronal alterations. I have further comments aiming to improve the current manuscript:

- The authors performed additional interesting analyses of the BA-NAc fibre photometry recordings Fig. Sup 2b. The authors need to explain why the number of animals reported per group and trial changes across the behavioral paradigm. In Fig. Sup 2c, it is unclear if the BA-NAc signal amplitude differs between CON and CSS mice. The authors should perform similar analyses as presented in main Fig. 2g and compare the BA-NAc neural activity between Trials 11-20 with DS response and Trials 11-20 without DS response within each group.

- While agreeing with the authors that using z-scores may normalize signal variabilities between subjects to some extent, validation and quantification of Cre-mCherry co-staining with GCaMP6-eGFP are still required to strengthen the current data set. Considering the functional heterogeneity between the NAc core and shell, it is critical to ensure that the NAc viral injections were similar between animals. Further, "insufficient GCaMP6 signal" to justify the exclusion of animals from the studies is insufficient: what statistical analyses were performed to reject these animals, and in which conditions were the signals tested? The authors need to provide for the 48 animals tested in the current study the immunohistochemistry validation of NAc Cre-mCherry staining and respective BA GCaMP6-eGFP labeling.

Responses to Reviewers comments on manuscript 22-3899-B

Reviewers' comments:

Reviewer #2 (Remarks to the Author):

I appreciate the efforts that the authors try to address the concerns being raised. However, the major concern is still the novelty of this study. I don't think there is real conceptual advancement on the BA-NAc pathway involving stress-induced impairment of reward processing, although the authors emphasized chronic social stress here. Just as Reviewer 1 mentioned, the current study does not go deep enough to explore the underlying mechanism even in the revised version, such as how the specific reward and aversive neurons act and interact in CSS-induced impairment in reward processing. And I still doubt the necessity of retaining Fig 1, especially because of the fact is that Fig 2 has included all the messages shown in Fig 1. Application of transcriptome expression appears to be an interesting part of the study, but again there is no extended exploration therefore does not provide significant novelty. This result also appears to be dissociated from other parts.

We politely disagree with the Reviewer on the issue of novelty. The manuscript presents the first evidence for each of the following: (1) chronic stress leads to a reduction in reward learning associated with decreased activity in basal amygdala-nucleus accumbens (BA-NAc) neurons, potentially reward neurons; (2) chronic social stress leads to a reduction in reward-to-effort valuation associated with increased activity in BA-NAc neurons, potentially aversion neurons; (3) relating to (1), specific and chronic inhibition of BA-NAc neurons replicates the chronic stress effect on reward learning; (4) relating to (2), specific and chronic activation of BA-NAc neurons replicates the chronic stress effect on reward-to-effort valuation. Three complex and iterative experiments were necessary to yield this novel data set. That “*the current study does not go deep enough to explore the underlying mechanism*” of the novel inter-relationships identified, is because an appropriate exploration of the underlying mechanism – specifically, separate studies of BA-NAc reward neurons and aversion neurons – will require at least as many experiments as those reported on in the current manuscript. It would simply not be appropriate or realistic to include (1) the current novel evidence for the causal inter-relationships chronic stress – BA-NAc pathway neuronal activity – reward-directed behaviour, and (2) elucidation of the distinct functioning of reward-neuron and aversion-neuron BA-NAc pathways, within one manuscript.

Concerning retaining Figure 1 in the main manuscript: the Reviewer's important comments on the original version of the manuscript relating to absolute – but not relative – differences in the behavioural scores presented in Figures 1 and 2, further reinforced our opinion on the need to include both of these figures in the main manuscript.

Reviewer #3 (Remarks to the Author):

In the revised manuscript, Madur and collaborators have adequately revised the manuscript and addressed important questions regarding the interpretation of the analyses and results. Overall, the study, in its revised form, provides information to the research field of stress-related behavioral disorders and neuronal alterations.

Thank you very much, and we again thank the Reviewer for their very constructive suggestions for including some additional analysis at the first revision of the manuscript.

I have further comments aiming to improve the current manuscript:

- The authors performed additional interesting analyses of the BA-NAc fibre photometry recordings

Fig. Sup 2b. The authors need to explain why the number of animals reported per group and trial changes across the behavioral paradigm. In Fig. Sup 2c, it is unclear if the BA-NAc signal amplitude differs between CON and CSS mice. The authors should perform similar analyses as presented in main Fig. 2g and compare the BA-NAc neural activity between Trials 11-20 with DS response and Trials 11-20 without DS response within each group.

We think it will be clear to readers as to “*why the number of animals reported per group and trial changes across the behavioral paradigm*”: indeed, this is because for all subjects there are trials without responses (as explained clearly in the text and figures), the trials on which subjects do not respond do not follow any particular pattern, and therefore the number of animals per group and trial changes across the behavioural paradigm. If this was not clear to the Reviewer, then we understand, because this trial-by-trial analysis, as requested by the Reviewer at the first revision of the manuscript, constitutes a very fine/detailed level of data analysis and presentation.

In Fig. suppl. 2c, the absence of p values denotes that there was no significant difference in BA-NAc activity signal between CSS and CON mice. Reporting only statistically significant effects in the figures is the strategy that we have used throughout the manuscript.

As noted above, the additional analysis of the BA-NAc fibre photometry data included in the first revision has now resulted in the analysis already being very detailed, and we are of the opinion that, if we add any more, it will be counterproductive because readers will be overwhelmed by detail. Concerning statistical analysis of BA-NAc neural activity in trials with and without responses: In trials with DS-tone responses the mean DS-tone duration is 5-10 s in CON mice and 15-20 s in CSS mice, whereas in trials without DS-tone responses it is 30 s; accordingly, the duration of BA-NAc neural activity covers highly different time periods, particularly in the case of CON mice. Furthermore, per mouse, there are relatively few trials without a DS response compared with a DS response. Therefore, the validity of the statistical findings will be questionable. For each of the above reasons, we would prefer not to add further analyses that will risk making the overall analysis too complex.

- While agreeing with the authors that using z-scores may normalize signal variabilities between subjects to some extent, validation and quantification of Cre-mCherry co-staining with GCaMP6-eGFP are still required to strengthen the current data set. Considering the functional heterogeneity between the NAc core and shell, it is critical to ensure that the NAc viral injections were similar between animals. Further, “insufficient GCaMP6 signal” to justify the exclusion of animals from the studies is insufficient: what statistical analyses were performed to reject these animals, and in which conditions were the signals tested? The authors need to provide for the 48 animals tested in the current study the immunohistochemistry validation of NAc Cre-mCherry staining and respective BA GCaMP6-eGFP labeling.

We absolutely want to provide as much validation evidence as possible that the injection areas in the nucleus accumbens (NAc) and basal amygdala (BA) were highly similar within and between experimental groups. However, immunohistochemistry is not relevant to do this.

For “*co-staining*”, we would understand this to mean that the same region is being immunostained for (Cre-)mCherry and (GCaMP6-)eGFP. However, these fluorescent signals are only of interest independently: we used (Cre-)mCherry to allow estimation of the injection area of a retrograde AAV vector in the NAc, and (GCaMP6-)eGFP was used to indicate the distribution of GCaMP6 (expressed by a floxed-AAV vector) expressed by NAc-projecting BA glutamate neuron cell bodies in the BA. Without the presence of (Cre-)mCherry there will not be any (GCaMP6-)eGFP expression, and therefore there is no benefit to be had by investigating the coincident “*co-staining*” of mCherry and eGFP in the BA.

In order to visualize the fluorescent signals of mCherry and eGFP that enable, respectively, the estimation of (1) the injection area of the AAV retrograde-Cre-mCherry vector in the NAc and (2) the area of Cre-dependent expression by neurons of GCaMP6-eGFP vector injected into the basal amygdala, we do not need to conduct immunohistochemistry (i.e. amplification), because the native fluorescent protein signals are sufficient to allow microscopic imaging of the signal. This is already indicated in the current figures 4j, 5j, S1a, S4e, 7m, 9b, d.

Nonetheless, concerning the point underlying the Reviewer's comment, we absolutely agree that it is important to provide as much information as possible on the injection area for the NAc. As already stated in the Methods, Stereotactic surgery (pp. 30-31): "The NAc coordinates were selected based on those used in some previous studies (Namburi et al., 2015; Kim et al., 2016) and to target BA projectors to both NAc core and shell sub-regions." That is, the retrograde-Cre vector was injected using coordinates that targeted the NAc core/boundary of core and shell sub-regions, in line with the previous studies of the BA-NAc pathway that informed our study (e.g. Namburi et al., 2015; Kim et al., 2016). Accordingly, the following revisions have been added:

In Fig. S1, similarly to Fig. S1b (which shows the estimated BA location of the optic fibre tip and GCaMP6 expression), Fig. S1c has now been added showing the estimated expression area of the retrograde-Cre-mCherry vector in the NAc sub-regions: it shows that for most mice the majority of the expression area was in the NAc core and for most mice there was also some expression at the NAc core-shell boundary. Fig. S1c includes the data for all 39 mice that contributed fibre photometry and behavioural data to the experiment and statistical analysis, with CON and CSS mice indicated in different colours. It does not include the nine mice that were excluded due to one of: no/low GCaMP6 signal, misplaced optic fibre or statistical outlier (see below).

In addition, we have added supplementary figures showing the estimated expression area of the NAc core-shell retrograde-Cre-mCherry vector in the BA-NAc TeTxLC experiment (Fig. S5c) and the BA-NAc DREADDs experiment (Fig. S5f).

We have added sentences to the Discussion that (1) summarize the expression area in the NAc core/shell and the extent to which this was similar within and between experimental groups and experiments, and (2) indicate the importance of future study of the potential spatial separation of BA-NAc reward neurons and aversion neurons in NAc core and/or shell.

Discussion p. 14: "Based on histological findings, in the current study the majority of BA-NAc neurons studied projected to NAc core and the minority to NAc shell. As such, and similar to previous studies of BA-NAc neurons and reward-directed behaviour^{12,15,17}, it is not possible to attribute findings to BA projection to one particular NAc subregion; importantly, the core-shell distribution of the neurons studied was highly similar between groups within experiments and between experiments."

Discussion p. 19: (Simultaneously, it indicates the importance of molecular tools that will enable the targeted identification, manipulation and study of BA-NAc reward- and aversion-neuron subpopulations,) "and their respective projections to D1R- or D2R MSNs in NAc core or shell."

In the Methods and the Reporting Summary, we have now stated explicitly when the BA-NAc fibre photometry signal was tested and the subsequent decision was made on animal inclusion/exclusion. This is also indicated in Fig. 2b. The following text has been added:

Methods, Effects of CSS on BA-NAc glutamate neuron activity during reward-directed behaviour using fibre photometry, pp. 33-34: "On day 16 mice were placed in the conditioning chamber without any stimuli and connected to the patch cord: the GCaMP6 photometry signal of each mouse was recorded for 15 min to check for a sufficient and stable signal, and mice with no/low signal relative to background were removed from the experiment." "Five mice were excluded from behavioural-fibre photometry testing due to no or low GCaMP6 signal at signal testing on day 16 (CON n=3, CSS n= 2)."

Reporting summary: "In the CSS behaviour-fibre photometry experiment, 5 mice (10%) were excluded due to no/low GCaMP6 signal as assessed prior to onset of behavioural testing, 4 mice (8%) due to technical failure, 1 mouse (2%) due to a misplaced optic fibre, and 3 mice (6%) due to being statistical outliers."

REVIEWERS' COMMENTS:

Reviewer #3 (Remarks to the Author):

Madur and collaborators have improved their manuscript and also well addressed some potential concerns regarding the analyses, criteria of exclusion, and information on the viral strategies used in their study. The current manuscript will offer important information to the research field of reward learning, stress behavioral and neuronal adaptations, and stress-related behavioral disorders. I have no further comments.